# The Integration of Metabolomics with Other Omics: Insights into Understanding Prostate Cancer

**DOI:** 10.3390/metabo12060488

**Published:** 2022-05-27

**Authors:** Eleazer P. Resurreccion, Ka-wing Fong

**Affiliations:** 1Department of Toxicology and Cancer Biology, University of Kentucky, Lexington, KY 40506, USA; e.p.resurreccion@gmail.com; 2Markey Cancer Center, University of Kentucky, Lexington, KY 40506, USA

**Keywords:** multi-omics, metabolomics, proteomics, transcriptomics, genomics, prostate cancer

## Abstract

Our understanding of prostate cancer (PCa) has shifted from solely caused by a few genetic aberrations to a combination of complex biochemical dysregulations with the prostate metabolome at its core. The role of metabolomics in analyzing the pathophysiology of PCa is indispensable. However, to fully elucidate real-time complex dysregulation in prostate cells, an integrated approach based on metabolomics and other omics is warranted. Individually, genomics, transcriptomics, and proteomics are robust, but they are not enough to achieve a holistic view of PCa tumorigenesis. This review is the first of its kind to focus solely on the integration of metabolomics with multi-omic platforms in PCa research, including a detailed emphasis on the metabolomic profile of PCa. The authors intend to provide researchers in the field with a comprehensive knowledge base in PCa metabolomics and offer perspectives on overcoming limitations of the tool to guide future point-of-care applications.

## 1. Introduction

Metabolomics is the newest omic science in systems biology, following genomics, transcriptomics, and proteomics. The four omics are complementary in understanding the interrelated cellular functions of a specific disease phenotype [1]. Metabolomics is currently applied to various disciplines including environmental epidemiology, food technology, ecological restoration, and oncology. It is an analytical profiling technique that measures and compares large numbers of metabolites in a biological sample. Metabolomic analysis is performed to identify (untargeted, global, and top-down approach) and quantify (targeted, specific, and bottom-up approach) metabolites with the goal of understanding the mechanisms by which upstream molecules (genes, RNAs, and proteins) contribute to pathology [2,3,4]. It seeks to investigate how therapeutics affect treatment outcomes [5] by serving as biomarkers via the quantification of these small molecules [6,7]. Metabolites (≤1.5 kD), which include sugars, fatty acids, amino acids, nucleotides, alkaloids, and steroids [1,8] can sometimes be enzymatically transformed into epimetabolites, allowing them to regulate physiological processes [9,10,11]. Because biological matrices are complex with thousands of metabolites in them, the use of analytical methods such as metabolomics allows individual metabolite measurements to be managed [12,13,14,15,16,17,18]. The difference between the two types of metabolomics is that the untargeted approach identifies a single metabolite in a hypothesis-driven manner, while the targeted approach quantifies a metabolite of interest *a priori* [1,2,19]. In humans, the untargeted approach reveals functional changes to the metabolome as a result of endogenous (diet, exercise) and exogenous (environmental exposures, virus, and genotoxins) agents [20,21,22,23]. Because the untargeted approach deals with a vast number of unknown molecules with disparate physical and chemical characteristics, multiple protocols for sample preparation, data acquisition, and analysis are required including subsequent validation via the targeted approach [24]. Regardless, in both approaches, tools such as high-performance liquid chromatography (HPLC), mass spectrometry (MS), and nuclear magnetic resonance (NMR) are used to provide insights into the disease mechanisms [3,4,18,25]. For clarity, metabolism refers to the series of biochemical processes that generate energy via ATP, while the metabolome is the collection of metabolites that are produced by cells during metabolism. The number of metabolites (intermediates) in a metabolome depends on the biochemical pathway involved. Moreover, metabolomic and metabolic are distinct from each other in that the former refers to the actual omic approach while the latter is a term that signifies the relationship to metabolism. The process flow for LC-, MS-, and NMR-based metabolomic analysis for disease biomarker research is shown in Figure 1.

## 2. Metabolomics: The “Supra-Omic”

The four omic platforms can be applied complementarily in pathology; however, metabolomics shows remarkable advantages over genomics, transcriptomics, and proteomics. Despite being relatively new, its ‘supra-omic’ nature is due to its ability to provide a real-time snapshot of the physiological state of a cell, tissue, or organism because the measured metabolite concentration accurately reflects infinitesimal biochemical perturbations, both endogenous and exogenous. Metabolomics has advantages over the other omics. First, the metabolome is highly sensitive to functional cellular changes brought about by stimuli including diet, radiation, medications, and stress levels [27]. Metabolites are products or intermediates of a metabolic pathway and their measurement represents a direct and real-time functional readout of physiological status or cellular activity [6]. Second, metabolomic alterations are determined via multiple analyses of biofluids (urine, serum) and tissue extracts in vitro, tissues and organs in vitro, and tissues *operando.* Samples are conveniently obtained in clinical and point-of-care (POC) settings, making risk assessment, diagnosis, staging, and treatment response evaluation quicker and more accurate. Third, metabolomic procedures can easily be integrated into currently existing clinical infrastructure that utilizes established protocols for a timely, reproducible, and cheap results [28,29,30,31,32]. Fourth, data analysis in metabolomics is easier to handle than those for the other omics because only a small fraction of the human metabolome is associated with key dysregulated metabolic pathways in any disease. In contrast, there are tens of thousands of genes and proteins that are potentially linked to a disease, some of which are yet to be discovered [33,34,35,36,37,38,39]. The biochemical importance of various metabolites is still unknown; although, their number is still relatively small compared to the human genome (~19,000 to 22,000) [40,41,42]. The number includes both polar and non-polar metabolites, present in large (>1 µmol/L) or small (<1 nmol/L) concentrations [6,43]. Fifth, the other omics are partially effective in evaluating cellular functions because no previously defined correlation exists between gene/protein expressions and metabolism, considering that RNA can be spliced or undergo post-translational modification [42]. For example, only a small fraction of transcriptomic alterations correlates with changes in proteomic data [44,45,46]. Even alterations in both genome and proteome are hardly reflective of a diseased cell’s phenotype. However, recent clinical evidence suggests that mutated isocitrate dehydrogenase1/2 (IDH 1/2), the enzyme that converts isocitrate to α-ketoglutarate (αKG) in the tricarboxylic acid (TCA) cycle, causes the conversion of αKG to the oncometabolite D-2-hydroxyglutarate (2-HG), which is responsible for the epigenetic inhibition and cellular differentiation [47,48]. This development establishes the first direct link between gene mutation to metabolic activity and cellular function in hematologic malignancy, providing a promising clinical opportunity for targeting the oncogenic pathway via drugs. Metabolomics is not without challenges, particularly in the use of an untargeted approach and the limiting factor of identifying unknown metabolites [49,50]. Since the approach handles small and diverse metabolic precursors with varying physical and chemical characteristics at unsteady state concentrations, it is necessary to employ sophisticated experimental designs, sample preparations, imaging techniques, and analyses to capture the series of enzyme-mediated catalytic reactions. The other omic platforms typically utilize a single tool. However, metabolomics requires multiple steps [24,50]. Thus, metabolomics is labor intensive requiring excellent techniques; although, it still produces the most meaningful results in disease etiology thus far. Metabolomics in a clinical setting supports the identification of metabolic biomarkers for cancer detection and surveillance [24]. For example, high-resolution metabolomics was used to identify the top 5, 10, and 20 metabolites from plasma using HPLC coupled with a Q-Exactive high-resolution mass spectrometer [51]. The identification and analysis of high-frequency metabolomic biomarkers with tyrosine on top were reported in a review for breast cancer [51] and a recent study by a French cohort utilized untargeted metabolomics in breast cancer to predict disease outcome [52]. Figure 2 depicts the hierarchical interrelationships among the omics.

## 3. Integration of Metabolomics to Other Omic Platforms

The biological activity of metabolites is a systems biology issue [53,54,55,56,57]. Combining metabolomics with other omics is attractive because the integration elucidates networks of molecular mechanisms in tumorigenesis [58,59,60], and can enhance personalized medicine [61,62,63,64,65,66,67]. For instance, using combined MS and HPLC can obtain information about individual differences in a patient’s metabolome and proteome, something that is difficult to achieve by solely using next-generation sequencing (NGS). NGS strategies as diagnostic solutions can analyze protein-coding regions associated with a patient’s disease, but it is insufficient in terms of adequately predicting temporal cellular states. The integration of all data from these omics is critical insofar as to suitably apply personalized medicine [68] because a metabolite connects a downstream target to a specific annotated gene [69]. The target in turn influences the gene to form a feedback loop mechanism [22], as shown in Figure 2. The importance of integration is seen in some very recent coupled metabolomic–genomic research [70,71,72,73]. The association of metabolites in gene expression, transcription, and translation is more significant than acting as a data sink. The activity of metabolites and associated enzymes is controlled by transcription factors such as androgen receptors (AR) [74,75] or estrogen receptors (ER) [76,77,78,79,80]. For prostate cancer, AR signaling is critical in the growth of prostate tumors given that androgen is required in de novo lipogenesis [81]. This phenomenon enables the tumors to proliferate despite androgen deprivation therapy (ADT) because they generate steroids for sustained ATP production. In other instances, gene expression is controlled by metabolites [82,83,84,85,86,87]. Metabolites are active participants in enzymatic reactions [88,89,90,91,92,93] and they control protein and cellular functions [94,95,96,97,98] so they are essential in comprehensively characterizing disease pathogenesis [99,100,101,102,103,104]. This review article focuses on *the* metabolomic profile of prostate cancer (PCa) and the current state of metabolomics–diverse omics integration in PCa research. In the first part, current knowledge on biochemical pathway alterations in PCa is discussed, including advances in adapting PCa metabolomics. In the second part, progress in integrating PCa metabolomics with other omics is detailed. In the last part, future directions and concluding remarks are given. To the best of our knowledge, this review article is the first of its kind to focus on metabolomic-based multi-omic data integration. The authors intend to provide researchers in the field with a comprehensive knowledge base in PCa metabolomics applications.

## 4. Why Focus on Metabolomics for PCa Cancer Research?

The PCa metabolome contains metabolites that reflect the human body’s reaction to tumor progression. Differentiating the PCa metabolome from the general human metabolome is critical since the complex relationships among these metabolites and how they affect PCa development is still a fairly new research area. As previously mentioned, alterations in genome, transcriptome (blueprints), and proteome (execution) do not directly reflect phenotypic changes. However, accurate identification and measurement of relevant metabolites provide functional PCa information because they are the end products of complex biochemical reactions, which can sensitively monitor any internal and external DNA damaging agents, including environmental factors [105,106]. The number of human PCa metabolites currently being researched is still very low but these few metabolites are highly specific pertinent pathways [107]. PCa metabolites can be conveniently extracted from urine, plasma, blood, and tissue. Traditional human clinical metabolic studies on PCa rely on biofluids because they are convenient and non-invasive to extract. However, researchers and clinicians are moving toward extracting tissues since they are organ specific, which reflects localized biochemical perturbations [108]. Challenges associated with tissue metabolomics, however, involve invasiveness, low patient samples for robust biomarker discovery and validation, and non-standardized protocols for various types of tissues. In terms of the separation of hydrophilic and lipophilic metabolites, whether they are extracted from biofluids or tissue, one challenge is separation and resolution efficiency. Each sample matrix is different, based on source and individual. Thus, equipment in the metabolomic analysis must be optimized, calibrated, and re-optimized to ensure clear separation of the metabolite of interest. Moreover, different mixed solvent standards must be used either as single, combined, or biphasic solvents to ensure increased levels of detection for all biofluid, tissue, and cell line samples. Healthy prostate cells rely on glucose oxidation for ATP production, and they are characterized by low citrate metabolism within the TCA cycle resulting in citrate accumulation [109,110]. Malignant transformation of prostate cells, however, activates the TCA cycle by decreasing zinc levels and the cells rely heavily on lipids for energy [111]. Unlike other cancers, PCa cells are unique in that they are not glucose dependent (non-Warburg). These cells show higher levels of metabolites including choline and sarcosine and lower levels of polyamines and citrate compared to normal prostate epithelial cells [111]. 

Within the last 10 years, there has been a growing number of purely PCa metabolomic studies, exploiting the various instrumental platforms. Most of these studies focused on biomarkers discovery and therapeutic target identification [1,24,37,109,110,111,112,113,114,115,116,117]. In PCa, sarcosine and choline are the primary metabolites [118]. Urea cycle metabolites such as arginosuccinate, arginine, and proline are elevated in PCa than in benign controls [119]. The study found that the oncogenic pathways HIF1α and NFκB were positively correlated with fumarate levels, inducing low survival rates. The increased plasma concentration of sphingolipids and Cav-1 also positively correlates with PCa aggressiveness [120]. The study determined that Cav-1 alters cell lipid metabolism by increasing the catabolic conversion of sphingomyelins to ceramide derivatives, elevating synthesis and efflux of glycosphingolipid indicative of altered ceramide metabolism and scavenging of exogenous sphingolipid. The landmark study by Sreekumar and colleagues in 2009, although controversial, has garnered further confirmatory studies to validate its results [121]. Although uracil, kynurenine, glycerol-3-phosphate, leucine, and proline were slightly elevated, sarcosine was singularly increased in metastatic PCa, and a localized tumor compared to BPH. An immediate validation study that has conflicting results was from Jentzmik and colleagues [122]. Post-digital rectal exam (DRE) of 106 PCa patients and 33 control patients revealed that the creatine-normalized sarcosine level was not statistically different between the two cohorts, including the absence of correlation between biopsy of prostatectomy Gleason score. Subsequent confirmatory studies ensued, without a conclusion as to the validity of Sreekumar et al., 2009, or Jentzmik et al., 2010. However, in the Cao et al. study investigating sarcosine levels in urine supernatant and sediment, the creatine- and alanine-normalized sarcosine levels were statistically higher in PCa patients than in abnormal prostate without cancer patients or healthy patients, from both sample source and normalization protein [123]. A very recent study using a PCa urine-based ^1^H-NMR revealed that guanidinoacetate, phenylacetylglycine, and glycine were significantly increased while L-lactate and L-alanine were substantially decreased [124]. In the 20 metabolites identified, sarcosine was not even a player in PCa after employing principal component analysis (PCA), partial least squares-differential analysis (PLS-DA), ortho-PLS-DA (OPLS-DA), and the Wilcoxon test. Another conflicting result among most of these validation studies is that the knockdown of glycine-N-methyltransferase (GNMT), the glycine-producing sarcosine enzyme, inhibits PCa cell proliferation to further abolish malignancy via G1 cell cycle arrest and apoptosis in certain allelic frequencies and ethnicities, with only a few studies finding opposite conclusions [125,126,127,128,129,130,131]. Additionally, the metabolic differences between normal prostate and PCa cells were previously thought to be caused by androgen receptors and that ADT suppresses tumorigenesis. However, the emergence of castration-resistant PCa (androgen-independent) makes androgen targeting by drugs more complicated because of the unique PCa metabolic profile, pointing to the need to identify biomarkers for cancer screening via metabolomics [24,81]. These facts show that the link between current clinical practice and unexplored gaps in using metabolomics is still elusive considering that the metabolites found in various PCa research are non-harmonized and at times contradicting. However, despite limitations and future refinements in analytical technique, metabolomics is suitable and needed in PCa research.

## 5. Why Merge Metabolomics with Other Omics in PCa?

Merging metabolomics with the three omics provides a more comprehensive PCa analysis [3,132,133,134]. In PCa, integrated metabolomics is utilized in two fashions: individual omics are independently adapted, and their results are co-analyzed for correlation and pattern analysis using statistical means, and multiple omics are integrated into a single model, the results of which then represent a single biological phenomenon [135]. The first case is executed using the functional study approach in which multiple independently generated omics data are plotted into a known metabolic network [106,135]. This visual representation of data is a powerful tool, but the interpretation is subject to errors and bias. Another method is to compare a priori gene ontology (GO) terms to other metabolites genes, enzymes, or proteins that are shown to have differential expressions between normal cells and PCa [135]. Within the last decade, multiple studies have come out pairing metabolomics and genomics. For instance, the overexpression of phosphorylated oncogenes AKT1 and MYC were linked with phenotypic metabolic sets associated with defined metabolic pathways [136]. Information on metabolomic profiles and matched gene expressions provide insight into the function of the gene using gene-metabolite profiles [137]. Correspondingly, metabolites can determine a particular gene target that contributes to the gene annotations [22]. However, integrated metabolomics strategy requires high-throughput computational and mathematical techniques such as Bayesian models [138,139], deep learning models [140,141], and least square models [142,143]. A detailed review of the principles of analytical integration of metabolomics and multi-omics data was made by Jendoubi et al. [132]

Recent PCa studies (2018–2021) in integrated metabolomics and genomics have employed techniques such as LC- or GC- combined with MS, fluorometric assays, and seahorse flux analysis. A study investigated arginine starvation using CWR22Rv1, PC3, and MDA-MB-231 cell lines [144]. Results revealed that deficiency in arginine synthesis (defects in PCa), performed as arginine starvation, resulted in cell death via epigenetic silencing and metabolite depletion. cGAS-STING activation also contributed to cell death. Oxidative phosphorylation, DNA repair pathway, and Type I interferon response were dysregulated, contributing to a decrease in both arginine and αKG. In a 2020 study by Kim and colleagues, withaferin (WA) treatment in 22Rv1, LNCaP, and 22Rv1 for validation employed fluorometric-based metabolomics [145]. In all cell lines, mRNA and protein levels of key fatty acid synthesis enzymes were downregulated. Suppression of a acetyl-coA carboxylase, expression of fatty acid synthase, and PCa cell survival from WA treatment resulted in the expression of c-MYC, not AKT. Glyceraldehyde-3-phosphate (GA3P) and citrate were both decreased. The metabolite-PCa causality was investigated in a study that employed genome-wide association studies (GWAS) in metabolites related to lipid, fatty acid, and amino acid metabolism [146]. Thirty-five metabolites were associated with PCa, and 14 of those were found not to have causality with PCa progression. These research studies that identified key metabolites at the genomic level can then be used as therapeutic targets or directions for further research. 

Numerous integrated metabolomics and transcriptomics (2019–2021) have demonstrated the utility of a combined approach. A study in 2021 concluded that per- and polyfluoroalkyl substances (PFAS) exposure led to an increase in xenograft tumor growth and altered metabolic phenotype of PCa, particularly those associated with glucose metabolism via the Warburg effect, involving the transfer of acetyl groups into mitochondria and TCA (pyruvate) [147]. PFAS also increased PPAR signaling and histone acetylation in PCa. Using RWPE-1 and RWPE-kRAS samples and GC-MS, acetyl-coA and pyruvate dehydrogenase complex were both significantly altered. Chen and group evaluated EMT-PCa and epithelial PCa differentiation utilizing ARCaP_E_ and ARCaP_M_ samples in LC-MS and a glucose uptake assay analytical platform [148]. The levels of aspartate, glycolytic enzymes (except for glucose 2 transporters), pyruvate dehydrogenase kinase 1/2, pyruvate dehydrogenase 2, and glutaminase 1/2 were all increased, while succinate dehydrogenase and aconitase 2 were decreased. PCa cells undergoing epithelial–mesenchymal transition (EMT) showed low glucose consumption and glucose metabolism in ARCaP_E_ downregulated. Glucose metabolism in transcription factor- (TF) induced EMT models was also downregulated. ARCaP_M_ cells showed increased aspartate metabolism. The carnitine palmitoyl transferase I (CPT1A) expression was analyzed by a study using the LNCaP-C4-2 and UHPLC-MS platform [149]. Results showed that ER stress, serine biosynthesis, and lipid catabolism were all upregulated, including the overexpression of CPT1A, which showed increased SOD2 when subjected to low fatty acids and no androgen. The implication was that high lipid metabolism and low androgen response resulted in worse progression-free survival. The group of Marin de Mas et al. conducted an aldrin exposure analysis via gene–protein reaction (GPR) associations to determine the effects on carnitine shuttle and prostaglandin biosynthesis [150]. Nineteen metabolites were found to be both consuming and producing. The application of a novel stoichiometric gene–protein reaction (S-GPR) (imbedded in genome-scale metabolic models, GSMM) on the transcriptomic data of Aldrin-exposed DU145 PCa revealed increased metabolite use and production. Carnitine shuttle and prostaglandin biosynthesis were shown to be significantly altered in Aldrin-exposed DU145 PCa.

There was a total of four recent PCa investigations using integrated metabolomics and proteomics from 2019 to 2021. One of them analyzed mast cell (MC) and cancer-associated fibroblasts (CAF) in PCa tissues from prostatectomy patients [151]. Transcriptomic profiling of MCs isolated from prostate tumor region showed downregulated SAMD14 while proteomic profiling of HMC-1 demonstrated an overexpression of SAMD14. Modified SAMD14 protein was associated with immune regulation and ECM processes. The group of Blomme et al. characterized AR inhibition (ARI) using the wild, bicalutamide-, appalutamide-, and enzalutamide-resistant LNCaP cells via LTQ-OVMS, FT-MS, QEO-MS, and LC-MS [152]. 2,4-dienoyl-coA reductase (DECR1) knockout induced ER stress and stimulated CRPC cells to undergo ferroptosis. DECR1 deletion in vivo, on the other hand, inhibited lipid metabolism, and reduced CRPC tumor growth. Both glucose metabolism and fatty acid β-oxidation were altered. Li et al. analyzed the silencing of FUN14-domain-containing protein-1 (FUNDC1) in PC3, DU145, and C42B cell lines [153]. A decrease in levels of pyruvate, cis-aconitase, α-ketoglutarate, and succinate accompanied by an increase in levels of glutathione and ROS were observed. FUNDC1 was shown to affect cellular plasticity via sustaining oxidative phosphorylation, buffering ROS generation, and supporting cell proliferation. Lastly, the team of Dougan et al. conducted a knockdown of peroxidasin (PDXN) in RWPE1, DU145, PC3, 22Rv1, and LNCaP [154]. PXDN overexpression was positively correlated with PCa progression, while PXDN knockdown increased oxidative stress, ROS, and apoptosis.

## 6. Clinical Applications of Metabolomics in PCa

The metabolic signature of PCa is used in tumor diagnosis, staging, and continuous assessment of treatment outcomes. The fact that PCa is a metabolic disease makes it suitable for targeted therapeutics. Metabolomics opens tremendous avenues for improving clinical applications. Biomarker discovery is one of metabolomics’ clinical applications. Advances in imaging, such as magnetic resonance imaging (MRI), computed tomography, radionuclide scans, and positron emission tomography (PET), are capitalized for the accurate detection of PCa. Since PCa cells do not rely on the Warburg effect (aerobic glycolysis) like most cancer cells, they are therefore not addicted to glucose (non-glycolytic). Thus, it has low avidity to 2-[18F]-fluoro-2-deoxy-D-glucose positron emission tomography/computed tomography (FDG PET/CT) [155]. It is only in late-stage metastatic PCa does the Warburg effect manifest. Other F-labeled glucose tracers can be employed for glucose-independent PCa. During early-stage PCa, ATP is produced from lipids from androgen signaling to produce energy. In the case of ADT, they utilize de novo lipogenesis. OXPHOS is favored and aerobic glycolysis is downregulated, in contrast to other tumors wherein OXPHOS is evaded to prevent apoptosis. Such a shift is attributed to acidosis in the microenvironment (TME) [24,113,155,156]. FDG PET/CT can be used in this case. Another novel tracer in PCa diagnosis is the [18F]-fluciclovine or the anti–1-amino-3-18F-fluorocyclobutane-1-carboxylic acid [157,158]. Fluciclovine uptake by PCa cells via alanine-serine-cysteine transporter 2 differentiates non-prostatic neoplasms from metastatic PCa [157,159]. Suitable tracers can now be implemented with high diagnostic accuracy considering that this review paper detailed the metabolic differences among normal, benign, and metastatic PCa. Other than metabolic imaging, clinical samples can be directly analyzed using metabolomics. Surgically obtained samples of PCa and the surrounding normal tissues can now be compared using metabolomics. However, this method is least desirable for PCa screening and monitoring. For the purpose of PCa biomarker detection, biofluid samples are adequate. Although sarcosine was recently rejected as a valid PCa biomarker, new clinical evidence using metabolomics suggests that free amino acids such as ethanolamine, arginine, and branched-chain amino acids are potential biomarkers [160,161]. The second clinical application of metabolomics is in identifying PCa risk factors. PCa progression is rooted in oncogenic DNA mutations, such as germline mutations and somatic mutations. These DNA alterations are caused by risk factors including endogenous agents (diet, ROS, macrophage, and neutrophil) and exogenous environmental agents (radiation, metals, and chemicals). Exogenous agents directly interact with DNA while endogenous agents indirectly promote carcinogenesis by promoting TME conducive to mutation. Once damaged, the DNA causes altered metabolism through changes in chromatin accessibility, which in turn modifies the epigenetic landscape. These metabolic risk factors can be accurately determined via untargeted metabolomics in population cohort studies [24]. Lastly, metabolomics can be adapted in a clinical setting in the discovery of advanced therapeutics that target PCa metabolism. For example, a study analyzing AKT and MYC dysregulation in human normal and PCa samples revealed that dysregulation of AKT1 and MYC alters non-glucose-mediated pathways and their downstream targets [136]. Since MYC is one of the leading oncogenes in PCa development, it can serve as a potential drug target. Another study conducted on characterizing urine-enriched mRNA using BPH, PTT, normal, and PCa urine samples in UHPLC-HRMS revealed that glutamate metabolism and TCA aberration contributed to PCa phenotype via GOT1-mediated redox balance [162]. Alanine, aspartate, and glutamate metabolite levels were increased including the level of glutamic-oxaloacetic transaminase 1. GOT11 in this context is an appropriate therapeutic target. Metabolomics can also be combined with immunotherapy and single-cell sequencing to aid in the search for advanced PCa therapeutics [163]. A summary of all recent integrated metabolomic studies on cell lines and in clinical cohorts are summarized in Table 1, Table 2, Table 3 and Table 4.

## 7. Metabolomic Tools

The most prominent techniques in PCa metabolomics are chromatography coupled to MS (LC-MS and GC-MS) and NMR spectroscopy (mostly proton NMR, ^1^H-NMR) [5]. NMR is widely used in the screening of patient urine and blood plasma samples because it can be fully automated, reproducible, and metabolites are easily identified from simple one-dimensional spectra [32,164]. It does not require intensive sample preparation and separation, making it ideal to be paired with other tools [164]. However, it is difficult to quantify co-resonant metabolites and it has lower sensitivity compared to MS by up to 100-fold [27,32,165]. Regardless, NMR can detect temporal biochemical changes and monitor real-time alterations in metabolites before and after experimental treatment [32,165]. GC-MS method fractionates mixtures into metabolite components and then uses mass spectrometry to quantitate each metabolite [166]. However, it can only be used for volatile metabolites. It is cheap, reproducible, and has high sensitivity; although, sample preparation takes significant time [166,167]. An alternative to ^1^H-NMR and GC-MS is LC-MS, in which separation occurs in the liquid phase, which broadens its applicability. It is not time consuming and can identify and quantify hundreds of metabolites in a single extract [168,169]. However, it is costlier than GC-MS and is difficult to control potentially due to the ionization problems when in presence of other ions [168]. Separation using LC-MS can alter the metabolites’ molecular structure. Other PCa techniques include Raman spectroscopy, Fourier-transform infrared (FT-IR) spectrometry, thin-layer chromatography, and metabolite arrays [170,171,172,173].

In the subsequent sections, we will present the current state of knowledge on PCa research, utilizing metabolomics paired with genomics, transcriptomics, and proteomics. Herein, we queried PubMed using keywords such as “genomics, metabolomics, prostate cancer,” “transcriptomics, metabolomics, prostate cancer,” “proteomics, metabolomics, prostate cancer,” and “multi-omics, metabolomics, prostate cancer.” Accompanied by other database searches, we exhaustively compiled all paired and multi-omic studies employing metabolomics.

## 8. Metabolomics and Genomics

Heterogeneity in PCa tumors and their metastatic form makes functional impact assessment challenging [174,175]. Fundamental mutations in PCa involve tumor suppressors (inactivating mutations) and oncogenes (activating mutations) [176]. To better understand how metabolomic dysregulation and genetic alterations are related to PCa, the main drivers of PCa oncogenic activity must be elucidated: AR expression, PTEN locus mutation, p53 locus mutation, and c-MYC amplification. Detailed PCa genomic reviews were performed elsewhere [101,177]. This review focuses on paired genomic and metabolomic studies performed thus far.

*AR expression.* Aberrant changes in AR render it sensitive to androgen deprivation therapy (ADT) and AR pharmaco-antagonists (androgen insensitivity syndrome), two mainstream therapies in PCa [178,179]. Alterations in AR genes include point mutations and deletions. Mutations in the second zinc-finger ligand-binding domain of the AR receptor contribute to this insensitivity [176,180,181]. Repeated AR mutations have been associated with resistance to AR-targeted therapy in CRPC [176,180,181,182]. One notable tool used in analyzing AR-mediated biochemical pathways and target genes is ^13^C-glucose metabolic flux analysis [183,184]. In a study on AR-V7, which correlated to ADT resistance and poor prognosis, the authors intended to validate whether such resistance is caused by AR substitution or potential AR-V7-mediated downstream gene target modifications [185]. Results revealed that AR-V7 promotes PCa growth and enhances glycolysis as with AR, including high dependence on glutaminolysis and reductive carboxylation. However, confirmatory metabolomic flux assay revealed that the ensuing low citrate level in PCa is due to low consumption, not low synthesis [186]. Further, AR targets genes associated with enzymes active in aerobic respiration, fatty acid oxidation, and homeostasis [187,188,189]. Lipid metabolism is an AR-regulated pathway that affects the production of acetyl-coA and modifications in acetylation and glycosylation processes [190].

*PTEN locus mutation.* PTEN is a tumor suppressor, and the deletion of its gene at the 10q23 location inactivates its protein and lipid phosphatase activities. It is a regulator of the PI3KT/AKT pathway [176,191]. PTEN-deficient PCa cells such as LNCaP are targeted directly or indirectly to restore PTEN function, via the blockade of the PI3KT/AKT pathway in combination with chemotherapy and other drugs [192,193]. Subsequent studies have demonstrated a positive correlation between PTEN mutations and PCa aggressiveness [194,195]. In a recent study, PTEN loss was shown to be positively correlated with fatty acid synthetase (FASN) gene knockdown, the enzyme in de novo lipogenesis. The downregulation of both genes resulted in a decrease in stromal microinvasion [196]. Co-deletion of PTEN with other genes, such as PML1, promoted PCa tumorigenesis in mouse models and activated SREBP, a transcription factor that regulates de novo lipogenesis and adipogenesis [197].

*p53 locus mutation.* p53 is another tumor suppressor; mutations in its genes lead to PCa development and PCa treatment resistance [198,199]. p53 represses the expression of glucose transporters resulting in the inactivation of glycolysis and PCa cell glucose consumption. p53 expression promotes OXPHOS via the regulation of glutamine uptake via activation of glutaminase 2 (GLS2) [199,200,201]. p53 as a PCa tumor suppressor was first proven in a study linking p53 mutations in PCa cell lines and PCa primary human samples [176]. Consecutive p53 studies validated the functional role of p53 mutation, specifically via loss, on PCa progression [198,199,201]. In a recent study, phenethyl isothiocyanate (PEITC), a dietary compound, inhibits PCa cell growth by inducing apoptosis via rescuing mutant p53 in VCaP and LAPC-4 [202]. Loss in p53 is also associated with enhanced serine one-carbon glycine synthesis (SOG), responsible for DNA methylation [203].

*c-MYC amplification.* The proto-oncogene and regulator gene c-MYC is a transcription factor encoded by the MYC oncogene on 8q24, shown to be constitutively overexpressed in PCa [204,205,206]. Research indicates that c-MYC alters enzyme expressions associated with glycolytic pathways including HK2, PFK1, ENO1, LDHA, and GLUT1 concentrations [207]. Additionally, GLS1 and its associated transporters are regulated by c-MYC, thereby advancing glutamine metabolism [208]. Amplifying c-MYC activates the PI3K/AKT axis. A study demonstrated that in localized and metastatic PCa, there is a correlation between c-MYC amplification with PI3K-associated dysregulation, including PTEN and all AKT homologs [209]. Activities of c-MYC and AKT1 stimulate the increase in glycolytic and lipogenic-associated metabolites in all PCa cell models [210,211]. It is found that c-MYC expression is positively correlated with AR activity [212,213,214], as shown in a recent study [212]. However, in another study, c-MYC overexpression exhibited an antagonistic effect on AR activity and transcription in PCa cell lines due to both proteins co-occupying similar enhancer binding sites [215]. The AR target genes KLK3 (PSA) and GNMT were inversely correlated with c-MYC in advanced PCa [215].

In these paired approaches, genomic data preceded metabolomic data; although, it is unclear as to the time-sensitive effect of genetic aberration on downstream metabolite levels [105]. There has been an increase in metabolomic genome-wide association studies (GWAS) that seek to quantify the extent to which genetic manipulations affect metabolite levels. In humans, GWAS and exome sequencing revealed that genetic variations account for roughly 10–76% of metabolic aberrations in blood metabolome [216]. Chu et al. published an epidemiological-based multi-omic study [105] and Jendoubi et al. published a review article on metabolomics and multi-omics integration [132]. These papers focused on methodological paradigms non-specific to PCa pathology, which emphasizes computational/mathematical approaches. Our literature search within the last decade (2011–2021) resulted in 91 exclusive paired studies and was trimmed to 14 pertinent PCa studies. These are listed in Table 1.

**Table 1 metabolites-12-00488-t001:** Summary of genomic–metabolomic integration studies for PCa within the last decade (2011–2021) ^1,2^.

Reference	Experimental Condition	Sample/ *n* Samples	Analytical Tool for Metabolites	Altered Metabolites (+/−)	Dysregulated Metabolic Pathways	Main Findings
Hsu et al., 2021 [144]	Arginine starvation	Cell lines: CWR22Rv1, PC3, MDA-MB-231	LC-MS Seahorse flux analysis	Arginine metabolites (−) α-ketoglutarate (−)	Oxidative phosphorylation DNA repair pathway Type I interferon response	Deficiency in arginine synthesis (defects in PCa), performed as arginine starvation resulted in cell death via epigenetic silencing and metabolite depletion. cGAS-STING activation contributed to cell death.
Cai et al., 2020 [217]	Citrate synthase (CS) down- regulation	71 = adenocarcinoma 2 = leiomyo-sarcoma 1 = hyperplasia 6 = normal	UPHPLC-MS/MS Seahorse assay	Glyceraldehyde 3-phosphate (−) Citrate (−)	Lipid metabolism Mitochondrial function	CS expression: PCa > normal prostate. Decreased CS expression resulted in inhibited PCa proliferation, colony formation, migration, invasion, cell cycle in vitro, and low tumor growth in vivo. CS downregulation lowers lipid metabolism and mitochondrial function.
Kim et al., 2020 [145]	Withaferin (WA) treatment	22Rv1 LNCaP, 22Rv1 (validation) Hi-MYC	Fluorometric assay	ATP citrase lyase, acetyl-coA carboxylase 1, fatty acid synthase, carnitine palmitoyltransferase (−)	Fatty acid synthesis	WA treatment in all cell lines downregulated mRNA and protein levels of key fatty acid synthesis enzymes. Suppression of a acetyl-coA carboxylase, expression of fatty acid synthase, and PCa cell survival from WA treatment → expression of c-MYC, not AKT.
Adams et al., 2018 [146]	Metabolite-PCa causality	24,925 = GWAS metabolites 44,825 = GWAS PCa 27,904 control	Data mining and statistical analysis, no experimental tool	Lipids and lipoproteins Fatty acids and ratios Amino acids Fluids 35 metabolites association w/ PCa, 14 has no causality	Lipid metabolism Fatty acid metabolism Amino acid metabolism	35 metabolites were associated w/ PCa, and 14 of those were found not to have causality w/ PCa progression.
Khodayari-Moez et al., 2018 [136]	AKT and MYC dysregulation	60 = human PCa samples 16 = normal prostate	Data analysis, no experimental tool	Metabolites related to dysregulated metabolic pathways	D-glutamine and D-glutamate metabolism Fatty acid biosynthesis Fructose and mannose Metabolism Nitrogen metabolism Pyrimidine metabolism	Dysregulation of AKT1 and MYC alters non-glucose-mediated pathways and their downstream targets. MYC is one of the leading oncogenes in PCa development.
Heger et al., 2016 [128]	Sarcosine dehydro- genase (SDH) supplementation	PC3, LNCaP PCa murine xenograft (validation)	IEC	Glycine, serine, sarcosine (+) dimethylglycine and glycine-*N*-methyltransferase (slight +)	Sarcosine metabolism	SDH supplementation significantly increased levels of glycine, serine, and sarcosine, but slight increase in dimethylglycine and glycine-*N*-methyltransferase levels. PC-3 → 25, LNCaP → 32, overlapping → 18 differentially expressed genes.
Liu et al., 2015 [137]	Gene-metabolite association	16 = benign 12 = PCa 14 = metasta- sized	Mathematical, no experimental tool, second-hand LC/GC-MS from Sreekumar et al.	1353 genes 1489 metabolites	Non-applicable	Directed random walk global gene-metabolite graph (DRW-GM) = from integrated matched gene and matched metabolomic profiles →accurate evaluation of gene importance and pathway activities in PCa. Use of method in three independent datasets → accurate evaluation of risk pathways.
Shafi et al., 2015 [186]	Androgen receptor variant 7 (AR-V7)	LNCaP	Seahorse assay LC-MS	Glucose/fructose (−) 3-phosphoglycerate, 2-phosphoglycerate (−) Pyruvate (+) Citrate (−) α-ketoglutarate (+) Malate (−) Oxaloacetate (+) Glutamine (+) Citrate (−)	Glycolysis via extracellular acidification rate (ECAR) Glutamine metabolism via reductive carboxylation Tricarboxylic acid (TCA) cycle Glutaminolysis	AR-V7 stimulated growth, migration, and glycolysis measured by ECAR (extracellular acidification rate) similar to AR. AR → increase citrate, AR-V7 → reduce citrate mirroring metabolic shifts (castration-resistant PCa). AR-V7 is highly dependent on glutaminolysis and reductive carboxylation → produce metabolites consumed by TCA cycle.
Gilbert et al., 2014 [218]	SNPs of vitamin D-PCa association	1275 = PCa 2062 = healthy controls	MS	25-hydroxyvitamin-D (25(OH)D) 1,25-dihydroxyvitamin, (1,25(OH)_2_D)	25(OH)D synthesis 25(OH)D metabolism	Vitamin D-binding protein SNPs were associated with prostate cancer. Low 25(OH)D metabolism score was associated with high grade.
Zecchini et al., 2014 [219]	Beta-arrestin 1 (ARB1)	C4-2 786-O	1,2-^13^C2 glucose assay GC-MS	Succinate dehydrogenase Fumarate hydratase	Oxidative phosphorylation Aerobic glycolysis	ARB1 contributes to PCa metabolic shift via regulation of hypoxia-inducible factor 1A (HIF1A) transcription through regulation of succinate dehydrogenase and fumarate hydratase in normoxic conditions. ARB1 was directly linked in PCa as a promoter by altering metabolic pathways. Survival of PCa cells in harsh conditions due to ARB1.
Hong et al., 2013 [220]	Metabolic quantitative trait loci (mQTLs) via genome-wide association study (GWAS)	214 = PCa 188 = control 489 = PCa (replication)	UPLC-MS w/ XCMS	Caprolactam Glycerolphosphocholine 2,6-dimethylheptanoylcarnitine Glycerolphosphocholine Bilirubin C_9_H_14_O_na_ Glycerophospho-N-palmitoyl ethanolamine Stearoylcarnitine Glycochenodeoxycholic acid 3-glucuronide	Fatty acid β-oxidation via acyl-CoA dehydrogenase	Seven genes (PYROXD2, FADS1, PON1, CYP4F2, UGT1A8, ACADL, and LIPC) and their variants contributed significantly to trait variance for one or more metabolites. Enrichment of 6 genes was associated w/ increased ACAD activity. mQTL SNPs and mQTL-harboring genes over-represented in GWAS → implications in PCa.
Poisson et al., 2012 [221]	Gene expression mapping	402 = original 488 = replication	Statistical and mathematical, no experimental tool	Non-applicable	Non-applicable	Convert gene information to p-value weight via 4 enrichment tests and 4 weight functions. Used p weights on PCa metabolomic dataset. Disjoint pathways → higher capability to differentiate metabolites than enriched pathways.
Lu et al., 2011 [222]	Single-minded homolog 2 (SIM2) expression	PC3 LNCaP VCaP DU145	LC-MS-MS	38 dysregulated metabolites	PTEN signaling PI3K/AKT signaling Toll-like receptor signaling	Lenti-shRNA in PC3 → downregulates SIM2 gene and protein → affects key signaling and metabolic pathways.
Massie et al., 2011 [223]	AR regulatory effects	LNCaP	NMR 1,2-^13^C2 glucose assay GC-MS	Calcium/calmodulin-dependent protein kinase kinase 2 (CAMKK2)	Glycolysis via activating 5’ AMP-activated protein kinase (AMPK)- phosphofructokinase (PFK) signaling	AR regulates aerobic glycolysis and anabolism in PCa. CAMKK2, a direct AR target gene, regulates downstream metabolic processes. CAMKK2 is important in androgen-dependent and castration-resistant PCa.

^1^ The list is non-exhaustive, tabulated as of the writing of this review article. ^2^ Total of 91 queries trimmed down to 14 integrated genomic-metabolomic PCa studies.

## 9. Metabolomics and Transcriptomics

The PCa’s genome has limited somatic mutations, but its gene expression profiles, as recorded in the transcriptome, are varied in both localized and metastatic PCa. Integrating transcriptomic data with metabolomic data reveals levels of known and unknown metabolites indicative of genetic aberrations or protein/enzyme expression. Table 2 summarizes a comprehensive decade-long study on paired transcriptomics and metabolomics. We scoured the literature and found 17 relevant publications.

**Table 2 metabolites-12-00488-t002:** Summary of transcriptomic–metabolomic integration studies for PCa within the last decade (2011–2021) ^1,2^.

Reference	Experimental Condition	Sample/ *n* Samples	Analytical Tool for Metabolites	Altered Metabolites (+/−)	Dysregulated Metabolic Pathways	Main Findings
Imir et al., 2021 [147]	Perfluoroalkyl sulfonate (PFAS) exposure	RWPE-1 RWPE-kRAS	GC-MS	Acetyl-coA Pyruvate dehydrogenase complex (PDC)	Glycolysis via Warburg effect and transfer of acetyl group into mitochondria TCA cycle Threonine and 2-oxobutanoate degradation Phosphatidylethanol-amine biosynthesis Lysine degradation Pentose phosphate pathway (PPP)	PFAS exposure led to increase in xenograft tumor growth and altered metabolic phenotype of PCa, particularly those associated w/ glucose metabolism via the Warburg effect, involving the transfer of acetyl groups into mitochondria and TCA (pyruvate). PFAS increased PPAR signaling and histone acetylation in PCa.
Tilborg and Saccenti 2021 [224]	Gene expression-metabolic dysregulation relationships	14 metabolic data sets, one of those is for PCa. 7 = tissue PCa 7 = tissue normal	Statistical, no experimental tool	Out of 72 metabolites investigated in PCa, 0 significantly differentially abundant metabolites were found (*p_adj_* < 0.05)	No enriched or dysregulated pathways for PCa	Topological analysis of Gaussian networks → PCa more defined by genetic networks than metabolic ones. PCa-related metabolites were not significantly altered between controls and PCa samples.
Wang et al., 2021 [225]	Differential metabolites between PCa and BHP	41 = PCa 38 = BPH	GC-MS GC/Q-TOF-MS Multivariate and univariate statistical analysis	12 metabolites (+/−) including L-serine, myo-inositol, and decanoic acid	L-serine, myo-inositol, and decanoic acid metabolism	L-serine, myo-inositol, and decanoic acid → potential biomarkers for discriminating PCa from BHP. The 3 metabolites → increased area under the curve (AUC) of cPSA and tPSA from 0.542 and 0.592 to 0.781, respectively.
Gómez-Cebrián et al., 2020 [226]	Dysregulated PCa metabolic pathway mapping	73 using serum and urine	NMR	36 metabolites (+/−) including glucose, glycine, 1-methylnicotinamide	Energy metabolism Nucleotide synthesis	36 metabolic pathways were dysregulated in PCa based on Gleason score (GS) (low-GS (GS < 7), high-GS PCa (GS ≥ 7) groups). Levels of glucose, glycine, and 1-methylnicotinamide → significantly altered between Gleason groups.
Chen et al., 2020 [148]	EMT-PCa and epithelial PCa differentiation	ARCaP_E_ ARCaP_M_	LC-MS Glucose uptake assay	Aspartate (+) Glycolytic enzymes (+) except for glucose 2 transporter (−) TCA cycle: pyruvate dehydrogenase kinase 1/2, pyruvate dehydrogenase 2 (+) Succinate dehydrogenase A, aconitase 2 (−) Glutaminase 1/2 (+)	Glucose uptake Aspartate metabolism Glycolysis TCA cycle Glutamine–glutamate conversion	PCa cells undergoing epithelial-mesenchymal transition (EMT) showed low glucose consumption. Glucose metabolism in ARCaP_E_ downregulated. Glucose metabolism in transcription factor- (TF) induced EMT models downregulated. ARCaP_M_ cells showed increased aspartate metabolism.
Joshi et al., 2020 [149]	Carnitine palmitoyl transferase I (CPT1A) expression	LNCaP-C4-2	UPHLC-MS	Acyl-carnitines Mitochondrial reactive oxygen species Superoxide dismutase 2	ER stress Serine biosynthesis Lipid catabolism Androgen response	Upregulated pathways via transcriptomic analysis → ER stress, serine biosynthesis, lipid catabolism. Overexpressed (OE) of CPT1A showed increased SOD2 when subjected to low fatty acids and no androgen → better antioxidant defense w/ CPT1A OE. High lipid metabolism, low androgen response → worse progression-free survival.
Lee et al., 2020 [162]	Urine-enriched mRNA characteriza-tion	Urine: 20 = BPH 11 = PTT 20 = PCa 20 = normal 65 = PCa (validation)	UHPLC-HRMS	Alanine, aspartate, and glutamate (+) Glutamic-oxaloacetic transaminase 1 (+)	14 metabolic pathways including aminoacyl-tRNA biosynthesis TCA cycle Pyruvate metabolism Amino acid pathways	Integrated gene expression-metabolite signature analysis → glutamate metabolism and TCA aberration contributed to PCa phenotype via GOT1-mediated redox balance.
Marin de Mas et al., 2019 [150]	Aldrin exposure analysis via gene-protein-reactions (GPR) associations	DU145	Dataset processing, no experimental tool	19 metabolites, both consuming and producing	Carnitine shuttle Prostaglandin biosynthesis	The application of novel stoichiometric gene–protein reaction (S-GPR) (imbedded in genome-scale metabolic models, GSMM) on the transcriptomic data of Aldrin-exposed DU145 PCa revealed increased metabolite use/production. Carnitine shuttle and prostaglandin biosynthesis → significantly altered in Aldrin-exposed DU145 PCa.
Andersen et al., 2018 [227]	Differential genes and metabolites	158 tissue samples from 43 patients	HR-MAS MRS	23 metabolites differentially expressed between high RSG and low RSG, including spermine, taurine, scyllo-inositol, and citrate	Immunity and ECM remodeling DNA repair pathway Type I interferon signaling	High RSG (≥16%) was associated w/ PCa biochemical recurrence (BCR). These high reactive stromata → upregulated genes and metabolites involved in immune functions and ECM remodeling.
Shao et al., 2018 [228]	Metabolomics-RNA-seq analysis	Tissue: 21 = PCa 21 = normal 50 = PCa and normal each (validation)	GC-MS	Fumarate Malate Branched-chain amino acid (+) Glutaminase, glutamate dehydrogenase ½ (+) Pyruvate dehydrogenase (+)	TCA cycle BCAA degradation Glutamine catabolism Pyruvate catabolism	Fumarate and malate levels → highly correlated w/ Gleason score, tumor stage, and expression of genes involved in BCAA degradation. BCAA degradation, glutamine catabolism, and pyruvate catabolism replenished TCA cycle metabolites.
Al Khadi et al., 2017 [229]	Peripheral and transitional zone differentiation	20 PCa patients undergoing prostatectomy	Network-based integrative analysis, no experimental tool	23 metabolites (+) including fatty acid synthase (FC = 2.9) and ELOVL fatty acid elongase 2 (FC = 2.8)	15 KEGG pathways including de novo lipogenesis and fatty acid β-oxidation	RNA sequencing and high-throughput metabolic analyses (non-cancerous tissue, prostatectomy patients) → genes involved in de novo lipogenesis: peripheral > transitional. Peripheral zone induced lipo-rich priming → PCa oncogenesis.
Sandsmark et al., 2017 [230]	CWP, NCWP, EMT evaluation	129 1519 samples (validation)	HR-MAS MRS MRSI	Citrate (−) Spermine (−)	TCA cycle	Increased NCWP activation via Wnt5a/Fzd2 Wnt activation mode → common in PCa. NCWP activation is associated w/ high EMT expression and high Gleason score. NCWP-EMT → significant predictor of PCa metastasis and biochemical recurrence.
Ren et al., 2016 [231]	Paired approach for altered pathways determination	25 = PCa and adjacent non-cancerous tissues each 51 = PCa and 16 = BHP (validation)	LC-MS TOF-MS	Sphingosine (+) Sphingosine-1-phosphate receptor 2 (−) Choline, S-adenosylhomoserine, 5- methylthioadensine, S-adenosylmethionine, Nicotinamide mononucleotide, Nicotinamide adenine dinucleotide, and Nicotinamide adenine dinucleotide phosphate (+) Adenosine, uric acid (−)	Cysteine metabolism Methionine metabolism Nicotinamide adenine dinucleotide metabolism Hexosamine biosynthesis	Cysteine, methionine, and nicotinamide adenine dinucleotide metabolisms and hexosamine biosynthesis were aberrantly altered in PCT vs. ANT. Sphingosine was able to distinguish PCa from BHP cells for patients w/ low PSA levels. The loss of sphingosine-1-phosphate receptor 2 signaling → loss of TSG (oncogenic pathway).
Torrano et al., 2016 [232]	Peroxisome proliferator-activated receptor gamma coactivator 1-alpha (PGC1α) assessment	150 = PCa 29 = control LNCaP DU145 PC3	LCHR-MS Stable isotope 13C-U6-glucose labeling	PGC1α (−) PGC1β Histone deacetylase 1	PGC1α pathway Estrogen-related receptor α (ERRα) pathway	PGC1α was a co-regulator and inhibits PCa progression and metastasis. Its deletion in murine prostate epithelium confirmed the finding. PGC1α dictates PCa oncogenic metabolic wiring, and its tumor-suppressive ability was mediated by the ERRα pathway.
Zhang et al., 2016 [233]	*Angelica gigas* Nakai (AGN) evaluation	5 mice per group	UHPLC-MS-MS	11 metabolites (+) including glutathione disulfide and taurine 11 metabolites (−) including lysine, tyrosine, and lactate	Methionine-cysteine metabolism Purine metabolism Citrate metabolism	Dosing w/ AGN → detectable decursinol, little decursin decursinol angelate.
Cerasuolo et al., 2015 [234]	Neuro- Endocrine transdifferen-tiation	LNCaP	H-NMR, Mathematical modeling	Creatinine + phosphor-creatinine (+) Glycine (+) Proline (+) Alanine (+) Fatty acids (+) Phospholipids (+) Glutathione (+) Glutamine (+)	Glucose oxidation Arginine and proline metabolism Glycine, serine, and threonine metabolism Glutamine and glutamate metabolism Glutathione metabolism	Hormone-deprived LNCaP cells were transdifferentiated to non-malignant neuroendocrine phenotype. Initially, LNCaP cells dwindled, neuroendocrine-type cells proliferated → later, neuroendocrine-type cells sustained LNCaP cells making them androgen-independent.
Meller et al., 2015 [235]	Metabolites analysis	106 = PCa	GC-MS LC-MS MRM	Malignant vs. non-malignant: 156 metabolites (+) 17 metabolites (−) Gleason score: 11 metabolites (+) 4 metabolites (−) ERG translocation: 53 metabolites (+) 17 metabolites (−)	Fatty acid β-oxidation Sphingolipids metabolism Polyamines metabolism Cholesterol metabolism	Fatty acid β-oxidation and sphingolipids metabolism were dysregulated in PCa relative to non-malignant tumors. TMPRSS-ERG translocated was positively correlated (causality) w/ metabolites from PCa samples. Advanced PCA tumors exhibited increased cholesterol metabolism → energy storage.

^1^ The list is non-exhaustive, tabulated as of the writing of this review article. ^2^ Total of 50 queries trimmed down to 17 integrated transcriptomic–metabolomic PCa studies.

## 10. Metabolomics and Proteomics

The proteome’s phenotype is closest to the metabolome’s [105]. Kim et al. identified proteins encoded by 17,294 genes [236] and Schroeder estimated that there are about 80,000–400,000 since one gene can encode multiple proteins [237]. In PCa, proteomics is applied to determine proteasomal degradation and aberrant metabolic processes. Most PCa studies focused on protein profiles and protein expression aberrations resulting from localized or metastatic PCa. A proteome sample is separated into components via gel- and liquid-based approaches. The gel-based method includes gel electrophoresis while the liquid-based method involves LC or LC-MS [101]. Implementing proteomics is expensive so integrated proteomics–metabolomics study is limited in the literature compared to genomics–metabolomics or transcriptomics–metabolomics studies. However, recent mapping development of the proteome and the emergence of top-down proteomics have made its use more manageable [105]. The integration of proteomic and metabolomic data has been focused on profiling, pathway mapping, and association studies. For example, PCa versus normal prostate cell differentiation is achieved via proteomics–metabolomics. The approach analyzes dysregulation in lipid metabolism and increases in protein phosphorylation [238]. Advancement in computing enables the coupled approach to move beyond simple pathway mapping. Herein, we summarized seven integrated proteomic–metabolomic PCa studies, presented in Table 3, within the last decade (2011–2021). The list was extracted from 86 online queries from PubMed and multiple databases.

**Table 3 metabolites-12-00488-t003:** Summary of proteomic–metabolomic integration studies for PCa within the last decade (2011–2021) ^1,2^.

Reference	Experimental Condition	Sample/ *n* Samples	Analytical Tool for Metabolites	Altered Metabolites (+/−)	Dysregulated Metabolic Pathways	Main Findings
Kopylov et al., 2021 [239]	Schizophrenia-PCa association	52 = PCa	Q-TOF MS UPLC	Cer(d18:1/14:0) ^3^Cholesta-3,5-dien-7-one 1α,25-dihydroxy-19-nor-22-oxavitamin D312:0 Cholesteryl ester24-hydroxy-cholesterol11-*cis*-RetinolElaidolinoleic acid14-hydroxy palmitic acid12-amino-dodecanoic acidL-Leucine	Sphingolipid metabolism ^3^ CholestanoidSteroid biosynthesisSteroid biosynthesis Bile acid biosynthesis Retinol metabolism Linoleic acid metabolismFatty acid biosynthesisFatty acid biosynthesis Valine, leucine and isoleucine degradation	Proteomic and metabolic data → input to approach employing systems biology and one-dimensional convolutional neural network (1DCNN) machine learning. Systems biology + 1DCNN → efficiently discriminate between: Unrelated pathologies = 0.90 (SCZ and oncophenotypes) Oncophenotypes/gender specific diseases = 0.93 (PCa). 1DCNN → high efficiency in PCa diagnosis.
Shen et al., 2021 [240]	Laser-capture-micro-dissection (LCM) androgen quantification	16 = PCa	LC-SRM-MS	Androsterone ^4^ Androstenedione Dehydroepiandrosterone Testosterone	Interleukin signaling ^4^ IGF signaling NOTCH4 signaling Wnt signaling PDGF signaling Steroid metabolism ECM signaling, RAF/MAPK signaling by integrins	Coupled parallel LC-MS-based global proteomics and targeted metabolomics → ultrasensitive and robust quantification of androgen from low sample quantity. LC-MS-based method → robust and reliable protein quantification in LCM, including highly accurate profiling of stroma and epithelial LCM of PCa patients.
Teng et al., 2021 [151]	Mast cell (MC) and cancer-associated fibroblasts (CAF) profiling	PCa tissue from prostatectomy patients BPH-1 HMC-1		SAMD14 (+) ^5^	Immune signaling ECM processes	Transcriptomic profiling of MCs isolated from prostate tumor region → downregulated SAMD14. Proteomic profiling of HMC-1 → overexpression of SAMD14 → modified proteins associated w/ immune regulation and ECM processes. Add HMC-1-SAMD14+ medium to culture of (CAF + prostate epithelium) → reduced deposition and alignment of ECM generated by CAF; suppressed tumorigenic morphology of prostate epithelium.
Blomme et al., 2020 [152]	Androgen receptor inhibitor (ARI)-based LNCaP characterization	LNCaP WT ^6^ LNCaP bicalut-res LNCaP apalut-res LNCaP enzalut-res	LTQ-OVMS FT-MS QEO-MS LC-MS	Metabolites associated w/ glucose metabolism (citrate, acetyl-coA) and lipid metabolism (+) for DECR1 overexpression Dihydroxyacetone phosphate and glycerol 3-phosphate (−) for DECR1 knockout	Glucose metabolism Fatty acid β-oxidation	2,4-dienoyl-coA reductase (DECR1) knockout → induced ER stress, and stimulated CRPC cells to undergo ferroptosis. DECR1 deletion in vivo → inhibited lipid metabolism, and reduced CRPC tumor growth.
Felgueiras et al., 2020 [238]	PCa-normal prostate differentiation	Tissue: 8 = PCa 8 = normal	FT-IR	Polysaccharide and glycogen (−) Nucleic acid (+)	Lipid metabolism Protein phosphorylation	FT-IR (spectroscopic profiling) and antibody microarray (signaling proteins) → dysregulation in lipid metabolism and increased protein phosphorylation.
Li et al., 2020 [153]	FUN14-domain-containing protein-1 (FUNDC1) silencing	PC3 DU145 C42B	LC-MS UPHLC	AAA+ protease LonP1 Complex V (ATP synthase) TCA intermediates: pyruvate, cis-aconitase, α-ketoglutarate, succinate (−) Glutathione, ROS (+)	TCA cycle Oxidative phosphorylation	FUNDC1 affects cellular plasticity via sustaining oxidative phosphorylation, buffering ROS generation, and supporting cell proliferation. FUNDC1 expression → facilitated LonP1 proteostasis → preserved complex V function and decreased ROS generation.
Dougan et al., 2019 [154]	Peroxidasin (PXDN) knockdown	RWPE1 DU145 PC3 22Rv1 LNCaP	LC-MS-MS	Metabolites that prevent oxidative stress and promote nucleotide biosynthesis (−) (i.e., desirable to increase oxidative stress and decrease nucleotide biosynthesis → apoptosis of PCa cells)	Oxidative stress response Phagosome maturation Eukaryotic initiation factor 2 (eIF2) signaling Mitochondrial bioenergetics Gluconeogenesis I	Increased PXDN expression positively correlated w/ PCa progression. PXDN knockdown → increased oxidative stress and decreased nucleotide synthesis. PXDN knockdown → increased ROS → decreased cell viability, increased apoptosis. PXDN knockdown → decreased colony formation.

^1^ The list is non-exhaustive, tabulated as of the writing of this review article. ^2^ Total of 86 queries trimmed down to 7 integrated proteomic–metabolomic PCa studies. ^3^ Altered metabolite indicates corresponding dysregulated metabolic pathway. ^4^ Enumerated metabolites are presented for quantification purposes using the coupled parallel LC-MS-based global proteomics and targeted metabolomics of LCM. The associated potential biochemical pathways are also listed. These pathways are not dysregulated since there are no experimental conditions applied. ^5^ Tumor-suppressor gene whose protein counterpart potentially induces regulation in immune signaling and ECM processes. ^6^ LCaP cell lines: LNCaP WT = LNCaP wild type; LNCaP bicalut-res = LNCaP bicalutamide-resistant; LNCaP apalut-res = LNCaP apalutamide-resistant; LNCaP enzalut-res = LNCaP enzalutamide-resistant.

## 11. Integrated Omic Analysis

Thus far, there are numerous studies combining multiple types of omic approaches and data within the last decade; however, there are few investigations in the literature that have employed metabolomics with other multiple omics. The excellent review by Zhang et al. showed PCa studies with few metabolomic-based omic combinations [177]. An example of a three-tier approach was performed by Oberhuber et al., in which they analyzed the effects of the expression of the signal transducer and activator of transcription 3 (STAT3) on PCa tumor growth, metabolite level, and PCa-associated metabolic pathways [241]. With transcriptomics, the group determined that high STAT3 expression corresponded to downregulation in OXPHOS. Similarly, proteomics revealed that STAT3 expression inhibits OXPHOS-TCA cycle activity. Nonetheless, the upregulation of pyruvate dehydrogenase kinase 4 (PDK4), an enzyme that lowers metabolism by inhibiting pyruvate-to-acetyl-coA conversion, resulted in the suppression of tumor growth [241]. These and other metabolomic-based multi-omic integration PCa studies are summarized in Table 4. It is important to note that omic science has expanded into new forms including epigenomics, lipidomics, volatilomics, and phosphoproteomics.

**Table 4 metabolites-12-00488-t004:** Summary of metabolomic-based multi-omic integration studies for PCa within the last decade (2011–2021) ^1,2^.

Reference	Experimental Condition	Sample/ n Samples	Analytical Tool	Altered Metabolites (+/−)	dysregulated Metabolic Pathways	Combined Modality/Main Findings
Kiebish et al., 2020 [100]	PCa prognostic markers identification	382 pre-surgical serum samples from PCa patients 267 = training set (validation) 115 = testing set (validation)	MS-MS HILC-MS LC-MS GC-TOF-MS	1-methyladenosine (+)	Cholesterol metabolism	Proteomics + Lipidomics + Metabolomics: Linear regression + Bayesian method + multi-omics → Tenascin C (TNC) and Apolipoprotein A1V (Apo-AIV), 1-Methyladenosine (1-MA), and phosphatidic acid (PA) 18:0–22:0, AUC = 0.78 (OR (95% CI) = 6.56 (2.98–14.40), *P* < 0.05) → high differentiating ability w/ and w/o BCR.
Oberhuber et al., 2020 [241]	Signal transducer and activator of transcription 3 (STAT3) expression	84 = PCa from prostatectomy patients	LC-MS-MS LC-HRMS	Pyruvate dehydrogenase kinase 4 (+)	Oxidative phosphorylation TCA cycle Pyruvate oxidation	Transcriptomics + Proteomics + Metabolomics: High STAT3 expression → OXPHOS downregulated (Transcriptomics). High STAT3 expression → TCA cycle/OXPHOS downregulated (Proteomics). High PDK4 expression → inhibited PCa tumor growth.
Itkonen et al., 2019 [242]	Cyclin-dependent kinase 9 (CDK9) inhibition	LNCaP PC3	Seahorse metabolic flux analysis	Acyl-carnitines (+)	Oxidative phosphorylation ATP synthesis AMP-activated protein kinase (AMPK) phosphorylation	Lipidomics + Fluxomics + Metabolomics: CDK9 inhibition → acute metabolic stress in PCa cells. CDK9 inhibition → downregulated oxidative phosphorylation, ATP depletion, and sustained AMPK phosphorylation. CDK9 inhibition → increased levels of acyl-carnitines
Gao et al., 2019 [243]	LASCPC-01 and LNCaP differentiation	LASCPC-01 LNCaP	GC-TOF-MS LC-MS	25 metabolites altered from control Carnitine (−)	Glycolysis One-carbon metabolism	Transcriptomics + Lipidomics + Metabolomics: 62 genes upregulated in LSCPC-01, 112 genes upregulated in LNCaP (Transcriptomics). 25 genes significantly altered from control (Lipidomics + Metabolomics). LASCPC-01: high glycolytic rate, low-level triglycerides. LNCaP: high 1C metabolism rate, low carnitine.
Kregel et al., 2019 [244]	Bromodomain/ extraterminal (BET)- containing proteins (BRD2/3/4) inhibitor analysis	22RV1 LNCaP VCaP PC3 DU145	LC-MS	Polyunsaturated fatty acids (+) Thioredoxin-interacting protein Interferon regulatory transcription factor (−)	Cyclin-dependent kinase 9 inhibition CDK9 hyperphosporylation Polycomb repressive complex 2 activity	Proteomics + Lipidomics + Metabolomics: BET inhibitors: affected AR+ PCa (22RV1, LNCaP, VCaP) more than AR- PCa (PC3, DU145). BET inhibitors → disrupted AR and MYC signaling at concentrations: (BET) < (BET inhibitors) (Proteomics).
Zadra et al., 2019 [245]	Fatty acid synthase (FASN) suppression via IPI-9119	LNCaP 22RV1 HeK293T RWPE-1	UPLC-MS-MS LC-MS GC-MS ^14^C-labeling	91 of the 418 metabolites modulated Malonyl-coA carnitine (+) Carnitine palmitoyltransferase 1 (−)	De novo fatty acid synthesis and neutral lipid accumulation ER stress response signaling Amino acid synthesis TCA cycle Carbohydrate metabolism Nucleotide metabolism	Lipidomics + Metabolomics: IPI-9119, a selective inhibitor of FASN altered the PCa metabolome by inhibiting fatty acid oxidation via accumulating malonyl-coA carnitine. Malonyl-coA carnitine accumulation → inhibited carnitine palmitoyltransferase 1 → FAO suppression. FA synthesis suppression → inhibited AR and AR-V7 expression. IPI-9119 → induced ER stress, inhibited AR/AR-V7 translation.
Murphy et al., 2018 [246]	PCa biomarker identification	158 = PCa prostatectomy patients	LC-MS-MS Statistical modeling	13 glycosylation metabolites (+) including tetraantennary tetrasialylated structures and A3G3S3	Glycosylation	Genomics + Transcriptomics + Proteomics +Lipidomics + Metabolomics: Integration of data across 5 omic platforms from tissue and serum → single AUC value that better differentiates aggressive PCa from the indolent type compared to AUCs obtained from single omics.
Hansen et al., 2016 [247]	TMPRSS2-ERG expression	129 = PCa samples from 41 patients 40 = PCa samples from 40 patients	HR-MAS-MRSI	Out of 23 metabolites, citrate and spermine (−)	TCA cycle Nucleic acid synthesis Citrate metabolism Polyamines metabolism	Transcriptomics + Metabolomics: ERG_high_ = low citrate and spermine concentrations → increased PCa aggressiveness (Metabolomics). Metabolomic alterations for ERG_high_ vs. ERG_low_ → more pronounced in low Gleason samples → implication: potential risk stratification tool.

^1^ The list is non-exhaustive, tabulated as of the writing of this review article. ^2^ Total of 82 queries trimmed down to 8 metabolomic-based integrated multi-omic PCa studies.

## 12. Metabolomic Profile of Prostate Cancer

In the U.S., PCa incidence and mortality is around 270,000 and 35,000, respectively, by 2022 [248]. It is the second leading cancer death in American men and the fifth leading cancer death among men worldwide [109,248,249,250]. PCa cells undergo substantial metabolic changes that define their unique phenotype [110,251]. The primary driver of PCa development is genetic alterations, but neoplastic transformations can occur, which further supplies energy to tumors [1,111,252,253,254,255,256,257,258,259]. Metabolic reprogramming is one of the hallmarks of PCa development [260,261,262,263,264]. PCa cells, unlike other cancers, do not depend on aerobic glycolysis for ATP production [81,265]. Instead, they obtain energy primarily from lipids via the activation of the TCA cycle [188,261,266]. Only in advanced metastatic PCa do cells favor lactate production in the presence of oxygen [267,268,269]. Although PCa cells do not exhibit the Warburg effect, they still produce lactate, which aids in immune escape, cell mobility, angiogenesis, and PCa development [270,271]. In normal prostate, citrate is accumulated [81] with glucose as the main source of energy [272]. In PCa, citrate is decreased [273]. The decrease in citrate lowers NADH production [81,273]. As a result, PCa cells produce energy less efficiently [274,275,276,277,278]. The accretion of zinc in normal prostate inhibits m-aconitase (m-ACO), the enzyme that catalyzes the isomerization of citrate to isocitrate in the TCA cycle [81,273]. Zinc is key in prostate malignancy since it dictates the tumor’s metabolic and energy consumption preference, growth and proliferation, and invasiveness. Simultaneously reducing citrate levels and preventing zinc accumulation drives PCa progression and metastasis [273,279,280]. PCa tissues have low levels of spermine in the prostatic fluid [279,280], contributing to their aggressiveness [281,282]. They are characterized by high levels of taurine [1,283,284], choline [285,286,287], sarcosine [121,288], myo-inositol [1,283,284], and pyruvate kinase M2 [1,283,284]. Androgen is the primary driver of PCa via AR signaling. Non-metastatic PCa is androgen dependent, with AR affecting the one-carbon metabolism and other transcription factors in PCa-related catabolic pathways [289]. Metastatic PCa is androgen independent, able to resist ADT by switching from one steroid receptor to another [290,291]. Glucocorticoids are often used in conjunction with antiandrogen agents and their effects are dependent on glucocorticoid receptors (GR) [290,292]. Research efforts have aimed at increasing glucocorticoid metabolism and GR responsiveness via hexose-6-phosphate dehydrogenase as a means of reversing metastatic PCa cells’ resistance to ADT [290]. In the succeeding sections, the canonical pathways associated with PCa progression are discussed: glycolysis, OXPHOS via the TCA cycle, *de novo* lipogenesis, and glycogenesis/glycogenolysis. Pentose phosphate pathway (PPP) and amino acid metabolism are included as non-canonical pathways. The metabolic profile between normal prostate and PCa cells is shown in Figure 3.

### 12.1. Glycolysis

The metabolism of healthy prostate epithelial cells and acinar epithelial cells are regulated by glycolysis [293,294]. In normal prostate, pyruvate in cytosol enters the mitochondria to be converted into acetyl-coA. Because glucose oxidation is incomplete in normal prostate, the bioenergetic balance is lower than the glycolysis–TCA tandem. Citrate accumulates in normal prostate due to the action of zinc, which inhibits m-ACO [113,114,295,296]. In essence, m-ACO compromises TCA, lowers citrate oxidation, and amasses citrate (produced from glucose and aspartate) in mitochondria, cytosol, and prostatic fluid. To sustain the energy requirement of the compromised aerobic respiration, non-essential biochemical pathways are limited [189]. However, in PCa, glycolysis is upregulated and reprogrammed, providing ATP energy for tumor proliferation [297,298]. Early-stage PCa limits glycolysis but stimulates enhanced OXPHOS [113]. Nonetheless, when it becomes metastatic and castration resistant, glycolysis is reinforced, including de novo lipogenesis, [299] amino acid metabolism, and nucleic acid synthesis [300]. Both benign and metastatic PCa cells exhibit some form of Warburg effect because ATP comes from aerobic glycolysis, not OXPHOS [267,301]. In fact, early-stage PCa cells derive their ATPs from lipids and other biomolecules, and when the cells have metastasized into late-stage PCa cells, they become wholly glycolytic.

Under anaerobic conditions, glycolysis is favored, and very little pyruvate is presented to the aerobic mitochondria [267,294]. Regardless of oxygen availability, PCa cells favor glycolysis [114,267,271,294]. The Warburg effect was initially associated with dysfunction in mitochondria but is now associated with the cell’s quick consumption of glucose, even for those pathways that are outside of mitochondria [302]. Because of the disregard for OXPHOS, PCa cells produce less ATP, but they efficiently convert glucose into lipids, amino acids, or nucleotides [303]. Glycolysis is regulated by AMP-activated protein kinase (AMPK) [304], which in turn activates the mammalian target of rapamycin (mTOR) complex 1 (mTORC1) [305]. Mouse models have revealed that PI3K/AKT/mTOR signaling pathways cause PTEN-deprived tumorigenesis in PCa [297,306,307,308]. This loss in PTEN results in the activation of pyruvate kinase M-2 (PKM-2), a key enzyme in aerobic glycolysis [309]. Another correlation exists between PTEN/p53 loss and elevated levels of hexokinases (HK2). The increase in HK2 has been attributed to the deletion of PTEN and p53 tumor suppressor genes in mouse models [310,311,312]. PTEN loss is associated with the activation of the AKT/mTORC1/4EBP1 signaling pathway [297,306,307,308], while p53 deletion is caused by the inhibition of miR143 synthesis [313,314,315,316]. PTEN/p53-mediated HK2 overexpression drives aerobic glycolysis, which promotes PCa metastasis. Another gene implicated in PCa cells’ survival is 6-phosphofructo-2-kinase/fructose-2,6-biphosphatase 4 (PFKFB4). The gene has demonstrated control over glycolysis and its associated mRNA; it is higher in metastatic PCa than in the localized version [317].

### 12.2. OXPHOS via the TCA cycle

Like glycolysis, AMPK controls the TCA cycle and is triggered when there is not enough ATP produced (e.g., high levels of AMP/ADP) [305]. It is a heterotrimeric protein encoded by the 5′-AMP-activated protein kinase gene (PRKA). AMPK protects cells from ATP decrease by regulating ATP consumption pathways. AMPK1 controls PCa oncogenes with its association with PI3K, mTOR, and MAPK pathways [318]. Activation results in reducing anabolic processes to limit energy use; however, AMPK controls lipid homeostasis [319] and mitochondrial homeostasis [320].

Acetyl-coA is produced in the cytosol from the *β-*oxidation of free fatty acids, oxidation of pyruvate, deamination and oxidation of amino acids, and oxidation of ketone (acetoacetate and *β-*hydroxybutyrate) [109,321,322,323]. Although the Warburg effect is crucial to PCa, OXPHOS via TCA provides additional energy in tumorigenesis. The normal and benign prostate epithelium promotes citrate synthesis over citrate oxidation [324]. In PCa, zinc is lost, m-ACO activity is enhanced, and citrate oxidation is activated [298]. The process ensures efficient and fast ATP consumption [325,326]. Rapid energy consumption guarantees PCa cells survival despite the limited availability of acetyl co-A. The production of oxaloacetate is also elevated to ensure sustained citrate oxidation [327]. For both upregulated glycolysis and the TCA cycle, the levels of glucose, lactate, and citrate are monitored using ^13^C isotope labeling metabolomics.

There are two zinc transporters relevant to PCa: SLC39 protein (Zrt- and Irt-like proteins ZIP) and SLC30 protein (ZnT) [328]. ZIP increases zinc levels in the cytoplasm by importing extracellular and vesicular zinc, while ZnT exports zinc out of the cell of moves them into mitochondria or lysosomes [328,329,330]. ZIP1-ZIP4 proteins have been shown to be downregulated in PCa [331,332]. ZIP1 (encoded by SLC39A1) was found to be absent in the TRAMP PCa model and was lower in RWPE2 human tumorigenic cells compared to RWPE2 non-tumorigenic cells [298]. ZIP1 was shown as the major zinc transporter because it is expressed in LNCaP and PC-3 cell lines, proving that its absence in some PCa studies is not due to mutation but rather transport [328]. ZIP2 (encoded by SLC39A2) reabsorbs zinc from prostatic fluid and is shown to be significantly downregulated in PCa compared to normal or benign prostate [331,333,334]. ZIP3 (encoded by SLC39A3) acts similarly to ZIP2; its protein expression changes with zinc status, but its mRNA expression is unchanged indicating post-translational modification [328,332]. Mutations in SLC39A4 gene-encoding ZIP4 were shown to be related to *acrodermatitis enteropathica* and its expression is decreased [328,335,336]. The knockdown of both ZIP1 and ZIP4 contributes to cell invasiveness [332]. Similarly, the knockdown of ZnT-1 as per their function, increases cell proliferation [337,338]. High levels of zinc were shown to induce apoptosis because zinc activates caspase-9, caspase-3, the release of cytochrome c from mitochondria, and the cleavage of poly(ADP-ribose) polymerase [339]. Low levels of zinc, on the other hand, reduce p53 and p21 concentrations in the nucleus and have been connected to high levels of PKB/AKT and Mdm2 phosphorylation. The importance of zinc in the TCA cycle is further emphasized by studies that reduce PCa invasiveness by inhibiting aminopeptidase N activity [340]. Reducing zinc in PCa cells elevated the expression of cytokines responsible for metastasis [337,338]. Zinc inhibits the activity of NF-kB, a transcription factor that regulates genes associated with PCa metastasis as well as reducing expressions of MMP-9, IL-6, IL-8, and VEGF genes [337,341]. Besides m-ACO, high-throughput mass spectrometry has revealed high levels of TCA enzymes such as citrate synthase, fumarase, and malate dehydrogenase in PCa cells [342].

### 12.3. De Novo Lipogenesis

Apart from serving as energy storage and directing intracellular signaling, lipids guide tumorigenesis because alterations in lipid or choline metabolites have ramifications in PCa cell proliferation [273,343]. Cholesterol inside the lipid droplets found in the cytosol of PTEN-deprived PCa cells proves the relationship between tumor development and lipid metabolism [344]. A major metabolic reprogramming in PCa cells is the upregulation of lipid synthesis for cell membrane formation, cell signaling, and cellular proliferation [114]. Early-stage PCa is characterized by the expression of lipogenic enzymes, but late-stage aggressive PCa shows the buildup of phospholipids (phosphatidylcholine), cholesterol esters, and triglycerides [273]. The letter type can also ingest exogenous lipids for synthesis. It is also observed that in late-stage metastatic PCa, acetyl-CoA is produced from acetate using acetyl-CoA synthetase 2 instead of being generated from glucose and glutamine [343].

*Fatty acids.* Studies show that the generated fatty acids are deposited in PCa cells [345]. However, no accumulation of lipids is observed despite an increase in de novo lipogenesis [346]. This may be due to the equilibrium between lipogenesis (cell membrane synthesis) and lipid oxidation (energy for survival and growth), wherein the elevated rate of fatty acid synthesis supplies energy to PCa cells while concurrently oxidizing lipids [113,278]. Evidence of such equilibrium in PCa can be seen by the overexpression of α-methylacyl-CoA racemase (AMCR), an enzyme that catalyzes lipid oxidation [345]. PCa is characterized by the presence of PRKAB1 and PFKFB4 genes required for cell proliferation, proving that like glycolysis and the TCA cycle, lipogenesis is AMPK regulated [317]. More proof of lipogenesis reprogramming in PCa is the high levels of phosphocholine, phosphoethanolamine, and glycerophosphocholine, responsible for cell membrane reconstruction and cell proliferation [347]. Lipogenic enzymes are increased in PCa due to the activation of the oncogenic-signaling pathway PI3K/AKT [277] while fatty acid enzymes are also elevated due to nuclear localization of AKT [348]. AR regulates fatty acids and cholesterol synthesis enzymes used in lipogenesis [323,349,350]. However, the elevated expression of a lipogenesis transcription factor sterol regulatory element-binding protein-1 (SREBP-1) in PCa alters the expression of fatty acid synthase/fatty acids by serving as a transcription factor for AR in a feedback loop fashion [351]. SREBP-1 further activates lipogenesis by increasing the production of reactive oxygen species (ROS) and NADPH oxidase 5—two species that promote PCa cell proliferation [352]. Fatty acid synthase also reprograms androgen-dependent and castration-resistant AR^+^ PCa models; thus, it can serve as a target that can potentially affect tumor aggressiveness [353].

*Cholesterol.* The increase in cholesterol synthesis, especially in the PCa cell membrane, is accompanied by high levels of choline and creatine [354]. PCa growth is mediated by AR, which can be addressed via AR antagonists such as enzalutamide. In PCa, cholesterol homeostasis is perturbed, and lipogenesis is upregulated. Studies suggest that a high level of circulating cholesterol and active cholesteryl ester synthesis in the blood increases the risk for PCa development [355,356,357,358,359]; although, some studies indicate that lower LDL and lower total cholesterol are associated with PCa at the time of diagnosis [360]. Recent evidence suggests that modulation of cholesterol metabolism or inhibition of its biosynthesis enzymes potentially suppresses tumor proliferation and metastasis [358,361,362,363,364]. Correspondingly, cholesterol esterification enzyme sterol-o-acyl transferases (SOAT) 1/2 or acyl-coenzyme A: cholesterol acyltransferase (ACAT) 1/2 are associated with PCa proliferation and invasion [359,365,366,367,368]. The homeostasis transcription factor SREBP 1/2 accumulates cellular cholesterol by increasing uptake and synthesis while the liver X (LXR) receptor promotes efflux [369,370]. SREBP increases ROS generation and NADPH oxidase overexpression causing PCa cells’ invasion and proliferation [352]. Blocking the SREBP-regulated metabolic pathway using statins has shown anti-tumor activity and, consequently, lowers AR signaling, which also controls cholesterol enzyme synthesis [371,372]. The downregulation of SREBP mediated by the inactivation of the PI3K/AKT/mTOR pathway (i.e., increase PTEN signaling) inhibits cholesteryl ester accumulation and aberrant SREBP-dependent lipogenesis [344,373,374,375]. The activation of the PI3K/AKT signaling pathway activates MDM2 (inhibitor of tumor suppressor p53), inhibits apoptotic genes BAX and GSK3, downregulates cell survival gene BAD, and inhibits cell cycle progression genes p21 and p27 [376]. An example strategy consisting of coordinated lipogenesis and AR signaling blockade is the use of fatostatin, which not only inhibited cholesterol biosynthesis but also caused G_2_-M cell cycle arrest and apoptosis [245,377,378]. The inverse relationship between statin use and PCa antitumor action is exemplified by several studies [363,379,380,381,382,383], which target major oncogenic/metabolic pathways such as AR-AKT complex and molecular mediators such as MK167 and cMYC [380]. Synergism between PI3K/AKT/mTOR dysregulation and PTEN-p53 inhibition in lipogenesis causes the Warburg effect and promotes PCa aggressiveness.

### 12.4. Glycogenesis/Glycogenolysis

Specific to PCa, a study added R1881 to androgen dependent PC3 cell lines expressing AR (i.e., androgen abscission), determining that in 5 days, cells were reduced (via G1 cell cycle arrest) and glycogen content was increased up to five times [384]. In addition, G6P was increased three times and both GS and GP were increased two times, providing evidence of enhanced glycogenesis. Moreover, glycogenolysis was inhibited by subjecting LNCaP cells to GP inhibitor CP-91149 and further validated that cell growth was curtailed [384]. This combined approach in targeting the glycogenesis pathway has since proven an efficacious PCa therapy. The metabolic reprogramming effects of glycogenesis in PCa were validated in another study; although, the authors employed CCL39 lung fibroblasts [385]. The authors showed that under low O_2_ levels, HF1/2 induced glycogenesis, as evidenced by increased glycogen stores and increased PGM1′s mRNA and protein levels. The generated glycogen served as feed to glucose-starved cells (hypoxia preconditioned cells), allowing them to survive via glycogenolysis (i.e., glycogen as glucose substitute) [385]. Such results parallel the Schnier study in that in order to combat PCa cell growth, invasion, and proliferation, glycogenolysis must be terminated through pharmacologic targeting of its intermediates and enzymes. Further, the approach opens a potential to also halt glycogenesis by AR deprivation therapy, which invariably stops glycogen as an alternative food.

### 12.5. Pentose Phosphate Pathway

PPP is a parallel glucose-degrading mechanism to glycolysis, with an interlink through fructose-6-phosphate (F6P) and GA3P [386]. The interlink with glycolysis is seen after isomerization of ribulose-5-phosphate (R5P) using transketolase and transaldolase. PPP is controlled by G6PDH wherein studies have indicated that this enzyme is increased in PCa [387,388,389,390,391]. G6PDH, NAPDH, and ribose synthesis were all upregulated in PCa through the action of AR signaling [343,388]. Further, upregulation of G6PDH through mTOR increased AR flux within PPP, as evidenced by the removal of the G6PDH-AR regulation mechanism following rapamycin treatment. While PPP’s role in PCa is only beginning to be understood, the results demonstrate its significant role in tumorigenesis [388].

### 12.6. Amino Acid Metabolism

Recent investigations have elucidated the role of amino acids in cancer metabolism [160]. Because the basis for amino acid metabolism is the generation of intermediates for the synthesis of nucleobases required for growing cells [392], depriving PCa cells with these intermediates can serve as PCa therapy [393]. Amino acids, like glucose, also fuel PCa progression. Glutamine, for example, is an important amino acid in human plasma shown to be associated with PCa [81]. It has an anaplerotic function in the human metabolism because it supports the TCA cycle by being transformed into glutamate and then to the intermediate α-ketoglutarate (glutaminolysis) [81,110,160,268]. PCa cells proliferate by maintaining glutamine metabolism through upregulating the glutamine transporter ASCT2 (encoded by the gene SLC1A5) and glutaminase, the enzyme in glutamine-glutamate conversion [394,395]. Glutamine in PCa is also responsible for acetyl-coA production; nitrogen donor for protein, nucleotide, lipid synthesis, and lipogenesis [110,113]. Glutamate is used in glutathione synthesis, which protects the cell from stress and PCa cell oxidation [396]. The two-prong role of glutamine in sustaining lipogenesis and glutaminolysis in PCa is highlighted in studies where both glutamine and the glutaminase transporter are overexpressed in tumor cells [397,398,399]. Whereas citrate is generated from OXPHOS via the TCA cycle, the same citrate is produced from α-ketoglutarate via the reverse TCA cycle (reductive carboxylation) [294,400]. This process supports the pathogenesis of PCa and hypoxia-inducible factor 1 (HIF-1) regulatory pathway because the glucose is rechanneled to the acetyl-coA pathway by the influx of glutamine [294]. α-ketoglutarate transformation (with CO_2_) essentially redirects the TCA cycle by producing isocitrate and citrate. The resulting citrate is transported into the cytosol, part of which is converted to acetyl-coA to support lipogenesis in PCa. The other part is then recycled as isocitrate in the TCA cycle [400]. The lactate (and some pyruvate) generated from reductive decarboxylation are consumed by PCa cells for their proliferation and anabolism [109].

Other crucial amino acids in the pathogenesis of PCa are serine (2-amino-3-hydroxypropanoic acid), glycine (aminoethanoic acid), proline (pyrrolidine-2-carboxylic acid), arginine (2-amino-5-guanidinopentanoic acid), leucine (2-amino-4-methylpentanoic acid), and sarcosine [(2-methylamino)acetic acid], among others [300]. Similarly, glutamine and proline are produced by PCa cells from arginine. While arginine is attributed to nitric oxide (NO) production, PCa cells appear to have lost their ability to synthesize arginine due to a deficiency in arginine synthetase [401,402]. Proline is also an important amino acid in that it maintains the level of pyridine nucleotides. Proline biosynthesis and its accompanying enzyme levels promote cancer cell growth, plasticity, and heterogeneity [403]. Another crucial amino acid in PCa is sarcosine, which was previously reported to be elevated in urine samples of PCa patients [121]. Sarcosine is an intermediate in glycine synthesis, produced from choline and methionine metabolism, and an essential component of glutathione, creatine, purines, and serine [404]. The sarcosine-glycine-methionine pathways promote purines and thymidylates synthesis, molecules that are essential in DNA synthesis and repair [405]. Amino acid synthesis in PCa tumor TME is also regulated by the AKT/mTORC1/4EBP1 signal transduction axis, which simultaneously loses PTEN and p53, resulting in HK2-mediated aerobic glycolysis—an event favorable to PCa proliferation, as seen in mouse models [113].

PCa cells are shown to increase the uptake of amino acids [160]. These amino acids are transported across cell membranes using mostly non-specific hydrophilic transporters. The most recognizable neutral and cationic amino acid transporter is the Na^+^- and Cl^−^-dependent SLC6A14. The L-type amino acid transporter 1 (LAT1, encoded by the SLC7A5 gene) is an antiporter, which imports branched-chain/high-molecular-weight amino acids (e.g., histidine, methionine, and phenylalanine) and thyroid hormones into the cells and exports glutamine and other essential amino acids [160]. SLC7A5 was shown in studies to be overexpressed in PCa cells [406,407]. LAT1 in PCa has a high affinity to leucine and it activates the mTOR signaling pathway [408,409]; thus, its inhibition results in tumor suppression. The dynamics between LAT1 and ASCT2 in PCa enable glutamine to enter the cytoplasm via ASCT2, glutamine to activate tumor-inducing pathways (i.e., glycolysis, TCA cycle), glutamine to leave the cytoplasm via LAT1, and leucine to enter the cytoplasm via LAT1 [160]. To summarize, Figure 4 presents an overview of the four dysregulated canonical pathways in PCa: glycolysis (Figure 4a), TCA cycle (Figure 4b), de novo lipogenesis (Figure 4c), and glycogenesis/glycogenolysis (Figure 4d).

## 13. Conclusions and Future Perspectives

The benefit of integrating metabolomics with other omics is possible with the advancement in metabolite quantification and imaging, allowing the discovery of clinically relevant biomarkers for precision medicine. The elegance of a multi-omic approach is its ability to elucidate multi-level real-time molecular interactions that reflect complex biochemical pathways and potential dysregulations. The approach is practical and has generated a tremendous amount of information within the last decade crucial to understanding PCa pathology. However, considerations must be made to effectively adapt an integrated metabolomics technique in a POC setting. First, while PCa genotyping and metabolic measurements are sufficiently robust to be translated into health care facilities, transcriptomics and proteomics still require solid quantification assays. Second, because the different omics develop at different rates, there needs to be data integration and harmonization from various domains. The use of a uniform ontology allows for a streamlined integration and interpretation of PCa omics data where they can be used for validation studies. Such integrated data must be high-quality with a high level of granularity and stored in a publicly available repository/databases. Third, one of the challenges in the PCa community, including other cancers, is the challenge of risk stratification based on survival results and clinicopathological indicators during PCa’s onset. This can be addressed by developing effective and precise therapeutic targets and biomarkers, which can only be achieved via an integrated omics analysis with metabolomics as its core. We are assured that this review provides comprehensive information on a metabolomics/multi-omics approach and its role in PCa.

## Figures and Tables

**Figure 1 metabolites-12-00488-f001:**
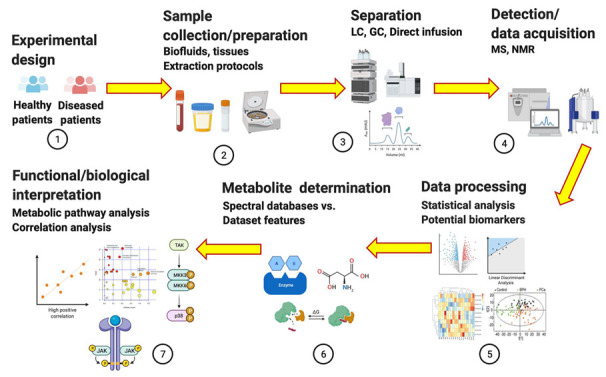
Process flow for untargeted and targeted metabolomics as applied to disease biomarker research. Figure drawn using BioRender [26].

**Figure 2 metabolites-12-00488-f002:**
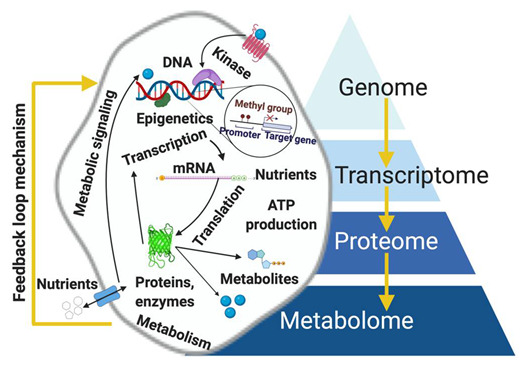
Hierarchical dimension of the omics reflecting metabolome in the most downstream position, directly linking genotype to the phenotype of a diseased cell. Results of metabolomics serve as inputs for further genomic analysis (i.e., feedback loop mechanism). Figure drawn using BioRender [26].

**Figure 3 metabolites-12-00488-f003:**
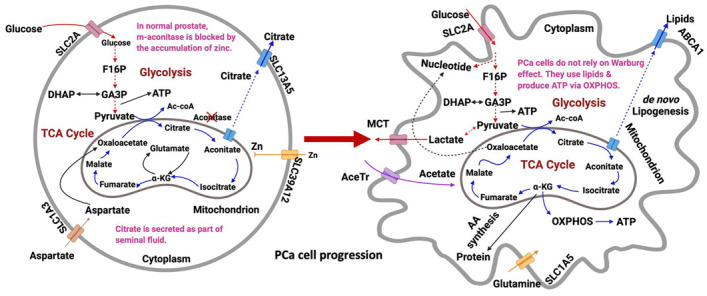
Metabolic profile of epithelial prostate cell during tumorigenesis. In the normal type (**left**), zinc inactivates m-aconitase (ACO), which accumulates citrate to prostatic fluid. In the malignant type, cells do not rely on the Warburg effect; although, they produce lactate. Instead, they consume lipids (generated via de novo lipogenesis), activate the TCA cycle, and stimulate OXPHOS for ATP generation. Enhanced glutamine metabolism and acetate consumption were also observed in PCa cells. Dashed lines indicate abridged pathways, and solid lines indicate direct pathways. Transporters for each species are indicated. Figure drawn using BioRender [26].

**Figure 4 metabolites-12-00488-f004:**
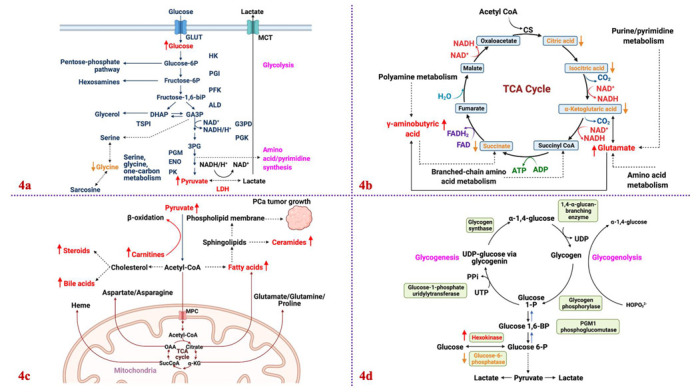
Schematic overview of the four canonical pathways dysregulated in human PCa tumorigenesis. Biofluid samples are extracted and analyzed to reflect changes in metabolites and enzymes for PCa biomarker discovery: Glycolysis (**a**), TCA cycle (**b**), de novo lipogenesis (**c**), and glycogenesis/glycogenolysis (**d**). Red font = increased metabolites/upregulated enzymes; Orange = decreased metabolites/downregulated enzymes. MCT = monocarboxylate transporter; HK = hexokinase; PGI = phosphoglucose isomerase; PFK = phosphofructokinase; ALD = aldolase; DHAP = dihydroxyacetone phosphate; GA3P = gyceraldehyde-3-phosphate; GA3PD = gyceraldehyde-3-phosphate dehydrogenase; PGK = phosphoglycerate kinase; 3PG = 3-phosphoglycerate; PGM = phosphoglyceromutase; ENO = enolase; PK = pyruvate kinase; LDH = lactate dehydrogenase; CS = citrate synthase; OAA = oxaloacetate; UDP = uridine diphosphate; UTP = uridine triphosphate. Figure drawn using BioRender [26].

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
