# Peer review of "The Integration of Metabolomics with Other Omics: Insights into Understanding Prostate Cancer"

_metabolites, 2022, doi:10.3390/metabo12060488_

Round 1

Reviewer 1 Report

Authors nicely presented the power of integration of metabolomics with other omics techniques. The review is a nice collection of relevant literature. I would point out couple minor concerns - 

1. I recommend to present the technical sections first and then the prostrate cancer metabolomics literature.

2. There are couple typos. For example, "de novo" is written in subscript in line 133. Please check for this type of errors troughout the manuscript. 

Author Response

General Comments

Authors nicely presented the power of integration of metabolomics with other omics techniques. The review is a nice collection of relevant literature. I would point out couple minor concerns. 

Response: Thank you.

Specific Comments

(1) I recommend presenting the technical sections first and then the prostrate cancer metabolomics literature.

Response: Done. We organized the manuscript such that the technical sections (Section 7: Metabolic Tools to Section 11: Integrated Omic Analysis precedes PCa metabolomics literature (Section 12: Metabolomic profile of prostate cancer).  

(2) Here are couple typos. For example, "de novo" is written in subscript in line 133. Please check for this type of errors throughout the manuscript. 

Response: Done. “De novo” in Line 133 was changed to normal script. Throughout the manuscript, punctuation errors were corrected, extra spaces were deleted, and minor sentence structures were rectified for fluency.

Reviewer 2 Report

The review article titled "The integration of metabolomics with other omics: Insights into understanding prostate cancer" came under the right time considering the importance of multi-omics approaches in discovering clinically relevant biomarkers for precision medicine. As author stated a multi-omic approach can elucidate multi-level real-time molecular interactions and aid in providing mechanistic insights to complex bio-chemical pathways and potential dysregulations; which is very important in understanding the biology of a disease. And the benefit of integrating metabolomics (which is close to the phenotype) with other omics reflects the above points, and is practically possible with the advancements in the field of metabolomics. 

Although the author has addressed this perspective in comprehensive understanding of the prostate cancer biology in the form of this well written review, it can be highly beneficial for the present scientists who are working towards using multi-omic approaches in understanding the biology of several diseases such as various other types of cancers, infectious diseases, etc., and in discovering and even validating effective and precise therapeutic targets and biomarkers for such diseases.

Overall, the review manuscript is well written, and can be published with the following minor revisions.

  1. Line 74 - remove extra period after the citation numbers.
  2. Line 312- correct the spelling continuous
  3. Line 318- correct the sentence starting with "since......
  4. Line 673- add a comma after the word "compiled" and check the sentence.
  5. Make the tables in a better way by adding borders, lines between columns, proper spacings, and fonts. 
  6. Check all the spacings, fonts, and grammar of the whole manuscript including the references thoroughly.

Author Response

General Comments

The review article titled "The integration of metabolomics with other omics: Insights into understanding prostate cancer" came under the right time considering the importance of multi-omics approaches in discovering clinically relevant biomarkers for precision medicine. As author stated a multi-omic approach can elucidate multi-level real-time molecular interactions and aid in providing mechanistic insights to complex bio-chemical pathways and potential dysregulations, which is very important in understanding the biology of a disease. And the benefit of integrating metabolomics (which is close to the phenotype) with other omics reflects the above points and is practically possible with the advancements in the field of metabolomics. 

Response: This review article is indeed timely and relevant. Thank you.

Although the author has addressed this perspective in comprehensive understanding of the prostate cancer biology in the form of this well written review, it can be highly beneficial for the present scientists who are working towards using multi-omic approaches in understanding the biology of several diseases such as various other types of cancers, infectious diseases, etc., and in discovering and even validating effective and precise therapeutic targets and biomarkers for such diseases.

Response: We agree. Despite this review’s focus on multi-omic application to PCa, information contained herein are translatable to other forms of cancer and disease.

Overall, the review manuscript is well written, and can be published with the following minor revisions.

Response: Thank you.

Specific Comments

(1) Line 74- remove extra period after the citation numbers.

Response: Done. All incidence of extra period after citation number were removed.

(2) Line 312- correct the spelling continuous.

Response: Done.

(3) Line 318- correct the sentence starting with "since......

Response: Done. The sentence was modified to “Since PCa cells do not rely on Warburg effect (aerobic glycolysis) like most cancer cells, they are therefore not addicted to glucose (non-glycolytic)”.

(4) Line 673- add a comma after the word "compiled" and check the sentence.

Response: Done. For clarity, the sentence was modified to “…we exhaustively compiled all paired and multi-omic studies employing metabolomics”.

(5) Make the tables in a better way by adding borders, lines between columns, proper spacings, and fonts. 

Response: Done. The tables were modified by adding borders and lines between columns. However, putting lines between columns in the final manuscript version is up to the Editor as per journal rules on tables. Spacings and fonts were also made uniform throughout the four tables.

(6) Check all the spacings, fonts, and grammar of the whole manuscript including the references thoroughly.

Response: Done. Spacings and fronts were made uniform throughout the manuscript. References were checked, verified, and updated.

Reviewer 3 Report

Dear Authors,

I read with interest the entitled “The integration of metabolomics with other omics: Insights into understanding prostate cancer”.

In this study the authors aimed to provide a comprehensive overview on metabolomics in PCa. First, I would strongly congratulate with Authors for their terrific effort and work. 

I found the present study interesting, well written and fluent to read – just minor English editing is needed (e.g., punctuation) -, concerning an actual topic. The title is descriptive of what authors have explored in their work. The background and scientific rationale for carrying out the study are well presented. Tables and Figures are clear and not repetitive, as well as the entire text. Subheadings are well presented, which make the text readable. The manuscript is adequately implemented with the relevant literature, yet, to give a complete overview on the topic, I would just suggest including also the interesting paper by Cerrato et al (DOI: 10.1016/j.aca.2021.338381), which provided pivotal data on PCa detection using untargeted metabolomics approach. I have no concerns or suggestions.

Author Response

General Comments

I read with interest the entitled “The integration of metabolomics with other omics: Insights into understanding prostate cancer”. In this study the authors aimed to provide a comprehensive overview on metabolomics in PCa. First, I would strongly congratulate with Authors for their terrific effort and work.

Response: Thank you. We are proud of this review, and we hope it helps scientists and clinicians in the field.

I found the present study interesting, well written and fluent to read – just minor English editing is needed (e.g., punctuation) -, concerning an actual topic. The title is descriptive of what authors have explored in their work. The background and scientific rationale for carrying out the study are well presented. Tables and Figures are clear and not repetitive, as well as the entire text. Subheadings are well presented, which make the text readable. The manuscript is adequately implemented with the relevant literature, yet, to give a complete overview on the topic.

Response: Thank you.

Specific Comments

(1) I would just suggest also including the interesting paper by Cerrato et al (DOI: 10.1016/j.aca.2021.338381), which provided pivotal data on PCa detection using untargeted metabolomics approach. I have no concerns or suggestions.

Response: Done. Cerrato et al. (2021) (Ref 50) was added to Section 2: Metabolomics: the “supra-omic”, particularly in the discussion related to untargeted metabolomics.

- “Metabolomics are not without challenges, particularly in the use of untargeted approach and the limiting factor of identifying unknown metabolites”.49,50

- “…untargeted metabolomics requires further validation for annotated metabolites via the targeted approach”.24,50

Reviewer 4 Report

The review article “The integration of metabolomics with other omics: Insights into understanding prostate cancer by Eleazer P Resurreccion and Ka Wing Fong, is a well written timely article on prostate cancer metabolomics. The review is focused on the benefits of integration of metabolomics with multi-omic platforms in prostate cancer. The review article provides an up-to-date on prostate cancer metabolomics and would be a good read for prostate cancer researchers.